# HIV-1 replication complexes accumulate in nuclear speckles and integrate into speckle-associated genomic domains

Ashwanth C. Francis [1✉], Mariana Marin[1], Parmit K. Singh[2,3], Vasudevan Achuthan[2,3], Mathew J. Prellberg[1], Kristina Palermino-Rowland[1], Shuiyun Lan[4], Philip R. Tedbury[4], Stefan G. Sarafianos[4], Alan N. Engelman[2,3] & Gregory B. Melikyan [1,5✉]

The early steps of HIV-1 infection, such as uncoating, reverse transcription, nuclear import, and transport to integration sites are incompletely understood. Here, we imaged nuclear entry and transport of HIV-1 replication complexes in cell lines, primary monocyte-derived macrophages (MDMs) and CD4+ T cells. We show that viral replication complexes traffic to and accumulate within nuclear speckles and that these steps precede the completion of viral DNA synthesis. HIV-1 transport to nuclear speckles is dependent on the interaction of the capsid proteins with host cleavage and polyadenylation specificity factor 6 (CPSF6), which is also required to stabilize the association of the viral replication complexes with nuclear speckles. Importantly, integration site analyses reveal a strong preference for HIV-1 to integrate into speckle-associated genomic domains. Collectively, our results demonstrate that nuclear speckles provide an architectural basis for nuclear homing of HIV-1 replication complexes and subsequent integration into associated genomic loci.

[1] Division of Infectious Diseases, Department of Pediatrics, Emory University School of Medicine, Atlanta, GA 30322, USA. [2] Department of Cancer Immunology and Virology, Dana-Farber Cancer Institute, Boston, MA 02115, USA. [3] Department of Medicine, Harvard Medical School, Boston, MA 02115, USA. [4] Laboratory of Biochemical Pharmacology, Department of Pediatrics, Emory University School of Medicine, Atlanta, GA 30322, USA. [5] Children's Healthcare of Atlanta, Atlanta, GA 30322, USA. ✉email: ashwanth.francis@emory.edu; gmelikian@emory.edu

The essential components of the HIV-1 ribonucleoprotein (RNP) complex include the viral RNA (vRNA), reverse transcriptase (RT) responsible for synthesis of viral DNA (vDNA), and integrase (IN) that catalyzes vDNA integration into host DNA. The RNP is encased in a protective shell that consists of ~1600 molecules of the capsid protein (CA) arranged into hexameric and pentameric rings[1] that form the viral core. Following HIV-1 fusion with the cell membrane, the core enters the cytoplasm where CA interacts with a number of host factors, including those that facilitate virus transport to the nucleus and nuclear import (reviewed in Refs. [2,3]). HIV-1 reverse transcription occurs within the confines of the reverse transcription complex (RTC), a large nucleoprotein complex that is derived from the viral core[4,5]. When the RTC becomes competent for integration, it is referred to as the pre-integration complex (PIC)[6]. Herein we use the term "viral replication complex" (VRC) to collectively denote the RNP, RTC, and PIC that are observed by virus particle imaging of infected cells.

HIV-1 VRCs enter the nucleus through the nuclear pore complex (NPC). Prior to nuclear entry, a portion of the capsid shell is believed to be shed through a poorly understood process termed uncoating[2]. The timely uncoating of the viral core is important for reverse transcription and VRC nuclear import[7,8]. Importantly, CA plays a key role in nuclear transport and preferential targeting of actively transcribed genes for HIV-1 integration[9–12]. Consistently, after uncoating at the NPC, a subset of CA molecules remains detectable in nuclear VRCs[13–19]. The cellular cleavage and polyadenylation specificity factor 6 (CPSF6), which interacts directly with CA[9,20,21], is a key mediator of intranuclear VRC trafficking. Disruption of CA–CPSF6 interaction results in VRC accumulation at the nuclear periphery[13,14,17,19], where HIV-1 integration occurs in heterochromatic lamina-associated domains (LADs)[19,22].

A wealth of information on the early steps of HIV-1 infection has been obtained using cell lines, including HeLa-derived cells (reviewed in Refs. [23,24]). By contrast, less information is available regarding the steps that lead to HIV-1 integration in primary monocyte-derived macrophages (MDMs). Infection of MDMs progresses comparatively slowly, likely due to delayed reverse transcription resulting from low deoxynucleotide triphosphate (dNTP) pools that are maintained by the dNTP triphosphohydrolase SAMHD1 (reviewed in Ref. [25]). In spite of the slow rate of reverse transcription, imaging of VRCs in MDMs has suggested that, similar to cell lines, loss of core integrity occurs within minutes after viral fusion[26]. In contrast, other imaging-based studies concluded that, whereas CA is largely lost upon VRC nuclear import in HeLa cells[13,15–17], little or no CA loss occurred in MDMs[14,15]. These discrepant results warrant further investigation of HIV-1 uncoating, nuclear import, transport, and integration in MDMs.

Here, by imaging nuclear entry of HIV-1 in living cells, we demonstrate that multiple VRCs traffic to and become compartmentalized within nuclear speckles (NSs) in MDMs. We find that NSs are preferred VRC destinations in multiple cell types and that HIV-1 transport is dependent on the interaction of CA with CPSF6. In turn, CPSF6 accumulates in HIV-containing NSs and helps maintain VRC compartmentalization. Importantly, integration site analysis reveals that HIV-1 has a strong bias for integration into NS-associated genomic domains (SPADs). Our results thus reveal the CA-dependent pathway for intranuclear HIV-1 transport and integration targeting.

## Results

### Multiple VRCs merge and form stable clusters in MDM nuclei.
We first examined the efficacy of HIV-1 pseudovirus infection in differentiated MDMs. Where indicated, we introduced Vpx to deplete MDMs of SAMHD1, using SIV virus-like particles (SIV VLPs) (Supplementary Fig. 1a). In agreement with the previous report[27], introduction of SIV Vpx resulted in SAMHD1 depletion and markedly improved HIV-1 infectivity compared to control cells treated with SIV VLPs lacking Vpx (Vpx(−)), Supplementary Fig. 1).

To visualize early stages of HIV-1 infection in MDMs, we used eGFP-expressing HIV-1 particles (HIVeGFP) pseudotyped with VSV-G that were co-labeled with integrase-mNeonGreen (INmNG) and the CA marker CypA-DsRed (CDR)[13]. The MDM nuclear envelope was labeled by expressing SNAP-Lamin (Supplementary Fig. 1a). Live-cell imaging of untreated or Vpx-treated MDMs was performed to visualize IN-labeled VRCs in nuclei over time ($n = 57$ Vpx(−) and 167 Vpx(+)). Single-particle tracking was utilized to trace nuclear VRCs back to respective entry sites at the nuclear membrane. A substantial lateral and axial movement of MDM nuclei greatly limited our ability to reliably track the nuclear entry step of the majority of VRCs. As a result, we were able to track nuclear envelope docking and uncoating for a subset (~30%) of VRCs that entered the nuclei of less mobile cells. For these particles, we consistently observed loss of CDR signal at the nuclear membrane, prior to nuclear import (Fig. 1a, b, Supplementary Fig. 2a, b and Supplementary Movies 1 and 2), similar to uncoating observed in HeLa-derived TZM-bl cells[13]. Interestingly, after entering the MDM nucleus, two or more IN-labeled complexes were observed to merge. Single particles either trafficked inside the nucleus toward each other and merged (Fig. 1a, b and Supplementary Movie 1) or moved toward and colocalized with an existing nuclear IN complex (Supplementary Fig. 2a, b and Supplementary Movie 2).

To directly demonstrate the merger of HIV-1 complexes in MDM nuclei, we co-infected untreated cells with two virus preparations labeled with either a green (INmNG) or red (INmCherry) IN marker. Infection with even a modest MOI of 1 (equivalent to <2% infection of Vpx(−) MDMs, Supplementary Fig. 1c, d) yielded over 78% of nuclear VRCs positive for both INmNG and INmCherry at 24 h post infection (hpi) (Fig. 1c, e), implying that VRC merger in MDM nuclei is a prevalent pathway. At later times after infection, all nuclear VRCs were double positive (e.g., Supplementary Movie 3).

Notably, preformed double-labeled VRCs in MDM nuclei could not be disrupted by treatment with high doses of CA-targeting antivirals or small molecule inhibitors of cellular functions such as transcription (Supplementary Fig. 3a and Movie 3), even upon prolonged incubation. Furthermore, VRCs positive for INmNG, INmCherry, and CA could be isolated from infected MDMs under stringent conditions, using detergent extraction and sonication (Supplementary Fig. 3b-d). Therefore, merged VRCs in MDM nuclei are stable physical entities and are not a consequence of diffraction limited fluorescence imaging. We designated these merged complexes as HIV-1 clusters.

HIV-1 complexes in MDM nuclei have been reported to contain vDNA and large amounts of CA/p24, leading to the suggestion that intact viral complexes enter the nuclei of these cells without shedding CA[14,28]. Given the strong tendency of nuclear VRCs to cluster, we sought to measure vDNA and CA signals in individual and merged nuclear complexes. vDNA was visualized by in situ labeling with the nucleoside analog 5-ethynyl-2'-deoxyuridine (EdU). Robust colocalization of EdU and INmNG signals was observed at 24 hpi (Supplementary Fig. 2c, d). The virtual absence of EdU signal in MDM nuclei in the absence of reverse transcription (nevirapine treatment, Supplementary Fig. 2e, f) demonstrated the lack of considerable DNA synthesis in these terminally differentiated cells. In parallel experiments, we

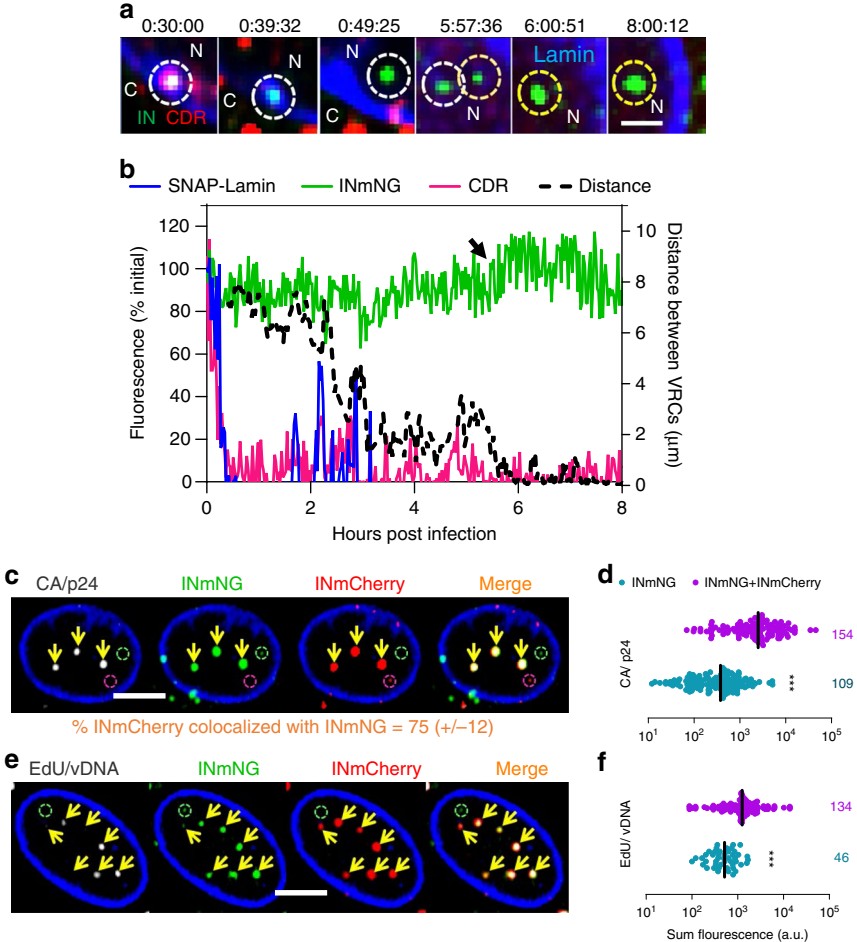

**Fig. 1 Multiple HIV-1 complexes traffic to distinct nuclear locations in MDMs. a, b** MDMs expressing SNAP-Lamin (blue) and infected at MOI 0.5 (corresponding to <30% infection of Vpx(+) treated cells after 5 days) with VSV-G pseudotyped HIV-1 co-labeled with INmNG and CDR were imaged at 90 s/frame consisting of 15 Z-stacks for 0.5–8 h. **a** Single Z-slice images with time stamps showing an IN/CDR co-labeled particle (white dashed circle) docked at the nuclear envelope that loses CDR and enters the nucleus. After nuclear entry of a second IN-labeled complex (orange dashed circle), the two complexes traffic toward each other, merge (yellow dashed circles) and stay together till the end of the experiment (see Supplementary Movie 1). **b** Mean fluorescence intensities of the single virus marked by white dashed circle in (**a**). The increase in INmNG signal at 6 h (arrow) represents merger of two nuclear IN complexes. Distance between the two IN complexes in (**a**) is plotted in the 2nd Y-axis (right). **c–f** Untreated MDMs were co-infected with HIV-1 labeled with INmNG or INmCherry markers at a total MOI of 1 for 24 h in the absence (**c, d**) or presence (**e, f**) of 5 μM EdU. Cells were fixed and immunostained for CA/p24 (**c, d**) or stained for EdU by click-labeling (**e, f**). **c, e** Images of a 2-μm-thick central Z-projection of the MDM nucleus showing merged (double positive) VRC clusters colocalized with (yellow arrows) CA (**c**) and EdU (**e**). Single-colored INmCherry or INmNG VRCs are marked by red and green dashed circles, respectively. **d, f** Fluorescence intensities of CA and EdU of double- vs. single-color INmNG puncta are shown. Scale bars are 1 μm in (**a**) and 5 μm in (**c, e**). Data in (**d, f**) are presented as median values ± SEM at 95% CI (confidence interval). N > 50 nuclei from three independent donors/experiments. The total number of IN puncta analyzed is shown on the right. Statistical significance in (**d, f**) was determined using a nonparametric Mann–Whitney rank-sum test, ***$p$ < 0.001. Source data are provided as a Source Data file.

compared the levels of CA associated with nuclear and cytoplasmic VRCs by immunostaining MDMs for p24. As a rule, brighter INmNG spots were strongly positive for CA/p24 and vDNA, whereas dimmer, presumably single VRC IN spots, often lacked detectable EdU and CA (e.g., Supplementary Fig. 2c, d). Similar results were obtained for HIV-1 co-labeled with INmNG and CDR as a marker for CA (Supplementary Fig. 2g, h), confirming that a subset of CA and CDR persists on VRCs beyond nuclear entry. A correlation between the INmNG signal and associated CA or CDR signal ($R^2 = 0.67$, Supplementary Fig. 2i) implied that both CA and CDR signals increase as a result of merger of multiple IN spots, in agreement with the live-cell imaging data (Fig. 1a, b and Supplementary Fig. 2a, b). This notion was further supported by the significantly higher CA and EdU signals associated with double-positive nuclear VRCs

compared to those associated with single-colored (INmNG or INmCherry) VRCs (Fig. 1d, f).

**HIV-1 accumulates in nuclear speckles of MDMs.** Our imaging results suggested that HIV-1 clusters may be confined to distinct nuclear compartments in MDMs and prompted investigation of VRC partitioning to NSs, which are enriched in actively transcribing genes[29–33] that are often targeted for HIV-1 integration[34]. We co-infected MDMs with INmNG- and INmCherry-labeled viruses and examined colocalization of VRCs with NSs. Immunostaining for markers of NSs, the serine and arginine rich splicing factor 2 (SRSF2, also known as SC35) or SON[33], revealed that nearly all dual-labeled VRC clusters (positive for INmNG and INmCherry) colocalized within NSs by 6 hpi (Fig. 2a and Supplementary Fig. 4a,

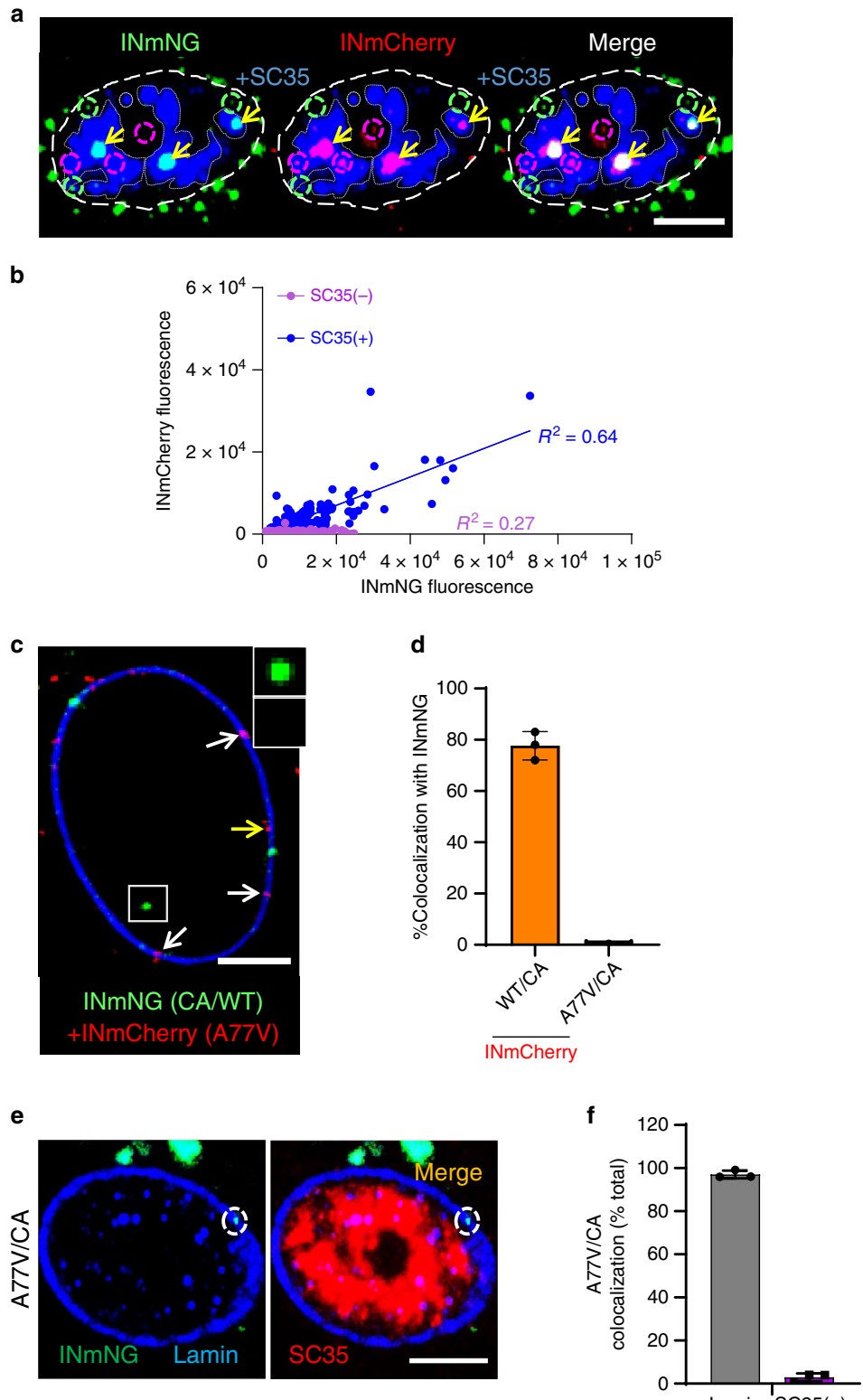

b). Consistent with the observed clustering of VRCs in NSs, the VRC-associated INmCherry signal in SC35-positive compartments correlated with INmNG signal, in contrast to nuclear VRCs residing outside of NSs (Fig. 2a, b). These results demonstrate that NSs are preferred destinations for HIV-1 complexes within MDM nuclei.

**CA is the determinant of HIV-1 transport to NSs.** The CA–CPSF6 interaction is critical for transporting HIV-1 complexes from the nuclear periphery into the nucleoplasm[13,14,17,19]. The A77V mutation in CA, which compromises the CA–CPSF6 interaction but only marginally impedes infection of MDMs[14,35], results in peripheral localization of nuclear HIV-1 complexes[14,19]. Accordingly, A77V CA mutant complexes labeled with INmCherry did not colocalize with intranuclear wild-type complexes labeled with INmNG following co-infection with both viruses (Fig. 2c, d) and failed to colocalize with NSs (Fig. 2e, f). These results implied

**Fig. 2 Multiple HIV-1 VRCs merge in nuclear speckles of MDMs. a, b** MDMs were co-infected with two VSV-G/HIVeGFP pseudoviruses labeled with INmNG or INmCherry (MOI 2), fixed at 6 hpi and immunostained for NSs (SC35). **a** A central section of MDM nucleus showing merger of INmNG and INmCherry VRCs in NSs (double positive IN clusters residing in NSs, yellow arrows). Single-labeled INmNG and INmCherry VRCs are marked with dashed green and red circles, respectively. NS contours are marked with semi-transparent gray dashed lines. **b** Fluorescence intensities associated with INmNG puncta inside (SC35(+)) or outside (SC35(−)) of NSs are plotted. **c–f** HIV-1 CA determines the nuclear penetration and speckle association of VRCs. **c, d** MDMs were co-infected with VSV-G-pseudotyped INmNG-labeled HIVeGFP virus bearing wild-type CA (green) and INmCherry-labeled pseudoviruses (MOI 2) containing either wild-type CA or the A77V CA (red). At 24 hpi, cells were fixed, immunostained for lamin, and colocalization of INmNG and INmCherry in the nucleus was quantified. **c** A central section of MDM nucleus shows nuclear INmNG spot for wild-type CA (white box) and nuclear membrane-associated INmCherry puncta of the A77V CA mutant (white arrows). The yellow arrows point to A77V CA INmCherry puncta on the nuclear side of Lamin. Inset shows the intranuclear WT CA INmNG spot (green) lacking A77V CA INmCherry signal. **d** The fraction of nuclear INmNG colocalizing with INmCherry puncta is shown. **e, f** MDMs were infected with INmNG-labeled VSV-G pseudotyped A77V CA mutant virus, fixed at 24 hpi, and immunostained for Lamin (blue) and SC35 (red). **e** Images show the A77V mutant INmNG puncta localized to the nuclear membrane failed to penetrate the nucleoplasm and reach SC35 compartments. **f** Quantification of the A77V CA mutant INmNG puncta colocalization with lamin or SC35(+) NS compartment. Scale bars in (**a, c, e**) are 5 μm and 0.5 μm in the inset (**c**). Error bars in (**d, f**) are SEM from n > 60 nuclei analyzed from three independent experiments. Source data are provided as a Source Data file.

that the CA–CPSF6 interaction, although largely dispensable for HIV-1 infection of MDMs[14,35], is the key mediator of VRC transport to NSs.

**NSs are preferred HIV-1 accumulation sites across cell types.** Analysis of INmNG-labeled nuclear VRCs in several cell lines including HEK293T, TZM-bl, Jurkat T cells, as well as primary CD4+ T cells, revealed that HIV-1 localized to NSs independent of cell type (Fig. 3a). By 6 hpi, the majority (>50%) of nuclear VRCs colocalized with SC35(+) compartments in all cell types. As observed with MDMs (Fig. 2b), NS-localized VRCs showed significantly enhanced INmNG signals compared to VRCs residing in the nucleoplasm (SC35-negative areas) (Fig. 3b). Interestingly, SC35 staining demonstrated a distinct morphology of NSs in MDMs compared to other cell types (Fig. 3a, c–e). On average, 15–45 SC35-positive compartments were detected in the nuclei of TZM-bl, HEK293T, Jurkat and primary CD4+ T cells, whereas only ~4 larger NSs were observed in MDMs (on average, 5- to 200-fold larger by volume compared to NSs in other cell types, Fig. 3a, d). At the same time, the fraction of nuclear volume occupied by all SC35-positive compartments was independent of cell type, approaching ~10% of the respective nuclear volumes (Fig. 3e). We also noted that the number or the size of NSs remained constant within the first 6 h of HIV-1 infection (Fig. 3c, d).

The apparent size and limited number of nuclear HIV-1 clusters in MDMs is in excellent agreement with the presence of only a few large NSs to which the viral complexes are transported (compare Fig. 2a, b, Supplementary Fig. 4a, b and Fig. 3). Collectively, our results imply that VRC trafficking to NSs is the prevalent pathway across diverse cell types.

**HIV-1 recruits CPSF6 to NSs.** We next analyzed colocalization of nuclear VRCs and endogenous CPSF6 in MDMs and TZM-bl cells. In agreement with prior work[14], CPSF6 exhibited diffuse nuclear staining in both cell types in the absence of infection, with no detectable accumulation in NSs (Fig. 4a, b). However, CPSF6 started to accumulate along with HIV-1 in NSs as early as 2–4 hpi (Fig. 4a, b and Supplementary Fig. 5a–d). Since both CPSF6 and INmNG (Supplementary Fig. 5a–d) tended to accumulate in NSs, we asked whether CPSF6 is enriched at sites of HIV-1 clustering. We accordingly normalized CPSF6 signal associated with VRCs to the respective INmNG signal (Supplementary Fig. 5a–d). This analysis revealed a >5-fold enrichment of CPSF6 at VRCs colocalized with NSs as compared to VRCs outside of NSs in both MDMs and TZM-bl cells (Fig. 4c, d).

Importantly, CPSF6 accumulation at NSs depended on the interaction with VRC-associated CA. A 30 min exposure to a comparatively high dose (25 μM) of the CA-targeting drug PF74 reduced VRC-associated CPSF6 signal in NSs to near background in MDMs and TZM-bl cells (Fig. 4a–d) without affecting the VRC signal (Fig. 4 and Supplementary Fig. 5c, d). PF74 also displaced CPSF6 from nuclear VRCs in other cell types (Supplementary Fig. 5e, f). Of note, PF74-induced CPSF6 displacement was not associated with a loss of CA signal from nuclear VRCs in TZM-bl cells (Supplementary Fig. 5g, h). On the contrary, 25 μM of PF74 significantly enhanced CA immunostaining of nuclear VRCs, perhaps owing to the exposure of CA epitopes after drug-mediated displacement of CA-interacting host factors in the nucleus (Supplementary Fig. 5g, h). These results imply that VRC-associated CA recruits CPSF6 to NSs in several cell types. Interestingly, a lower dose (2.5 μM) of PF74 failed to displace CPSF6 from NS-localized VRCs, even after prolonged (up to 5 days) treatment of MDMs, (Supplementary Fig. 5i, j). This is in contrast to the ability of this concentration of PF74 to effectively block nuclear import of HIV-1 in several cell types when added early during infection[13–15,36,37].

To gain further insights into the CPSF6/VRC interaction, we transiently expressed CPSF6 tagged at the amino terminus with a photoactivatable GFP (abbreviated PA-C6)[38] in TZM-bl cells. PA-C6 association with nuclear VRCs was visualized by live-cell imaging. Photoactivation of PA-C6 within a selected region of the nucleus of uninfected cells revealed that this protein was highly mobile (Fig. 5a, b and Supplementary Movie 4). In contrast, PA-C6 locally photoactivated at INmCherry-labeled nuclear VRCs remained stably associated with these complexes, unless displaced by treatment with 25 μM PF74 (Fig. 5c, d and Supplementary Movies 5 and 6). Thus, VRCs residing in NSs recruit and retain CPSF6 in a CA-dependent manner.

**Relocation outside of NSs diminishes nuclear VRC stability.** We next examined the effects of PF74-induced displacement of CPSF6 on the mobility of NS-associated VRCs. Prior to PF74 addition, nuclear VRCs exhibited a highly restricted motion, consistent with the very limited NS mobility[39]. The addition of 25 μM PF74 induced a quick and marked increase in mobility of nuclear VRCs in TZM-bl cells, promoting their long-range movements (Fig. 5e, f and Supplementary Movie 7). Importantly, increased VRC mobility was associated with their relocation outside of NSs, which was more pronounced in TZM-bl, HEK293T, and Jurkat cells (>3-fold) compared to primary CD4+ T cells (~2-fold, Fig. 5g, h). In contrast, this 30 min treatment was virtually without effect on VRC localization in MDMs (Fig. 5h).

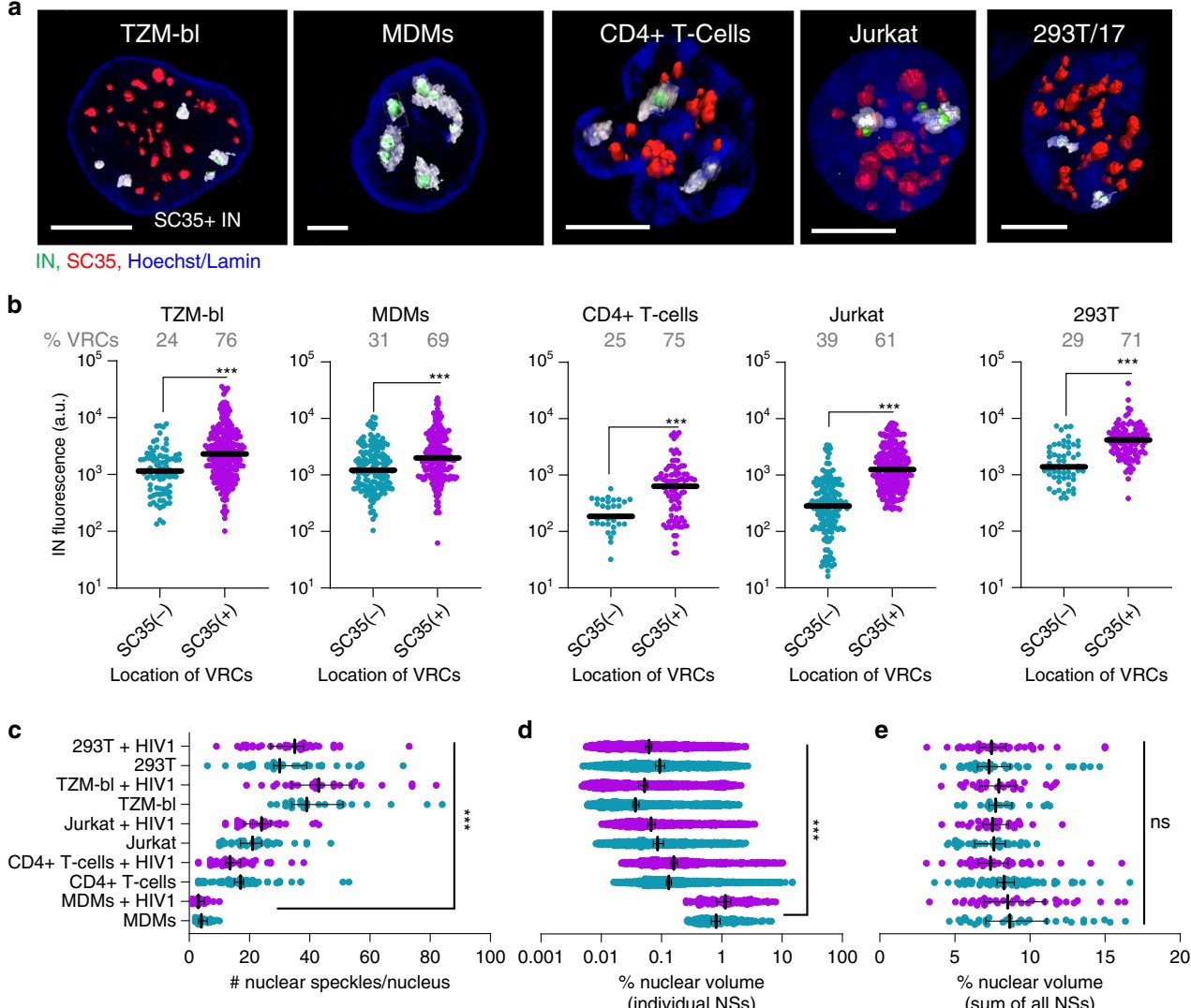

**Fig. 3 NSs are preferred HIV-1 accumulation sites in different cell types. a–e** MDMs, HEK293T, and TZM-bl cells were infected at MOI of 5 (determined on TZM-bl cells); Jurkat cells and primary CD4+ T cells were infected in suspension with the same virus supernatant and later adhered to a 8-well coverslip for microscopy analysis. All cell types were fixed at 6 hpi (uninfected cells were used as controls). Cells were immunostained for NSs (SC35) and lamin in TZM-bl and MDMs; the nuclei of T cells and HEK293T cells were stained with Hoechst-33342. NSs (SC35(+)) compartments were identified by three-dimensional image analysis. **a** 3D-rendered of images showing NSs (red), IN puncta (green), and nuclei (blue). NSs encompassing HIV-1 complexes are colored transparent gray. Scale bar: 5 μm. (**b**) Fluorescence intensities of IN puncta inside (SC35(+)) and outside (SC35(−)) nuclear speckles. **c** The number of NSs per nucleus in different cell types. **d** Percent of nuclear volume occupied by individual NSs. **e** Percent of nuclear volume occupied by all speckles. Infected cells (red symbols) are marked with cell type name +HIV-1. Data in (**b–e**) are presented as median values ± SEM at 95% CI. N > 50 nuclei for each cell type from >3 independent experiments in HEK293T and Jurkat cells, from three donors in primary CD4+ T cells, and >5 independent experiments in TZM-bl cells and MDMs. A total of 124 nuclear IN puncta were analyzed in CD4+ T cells and >200 IN puncta were analyzed in all other cell types (**b**). Statistical significance between uninfected MDMs and other cell types in (**b–e**) was determined using a nonparametric Mann–Whitney rank-sum test. ***$p < 0.001$. Differences in the number NSs between uninfected and HIV-infected cells in each cell type were insignificant, $p > 0.05$. Images in (**a**) are representatives of >120 nuclei from at least three independent experiments. Source data are provided as a Source Data file.

We next examined the fate of VRCs relocated from NSs to the nucleoplasm of TZM-bl cells by PF74 treatment and found that, importantly, exit from NSs was accompanied by disappearance of nuclear VRCs (Fig. 6a, b) and loss of infectivity (Fig. 6d). This treatment did not affect the CA signal associated with nuclear complexes that persisted after treatment (Fig. 6a), implying that the drug did not displace CA from VRCs. It should be noted that PF74 treatment that mediated VRC relocation from NSs and diminished the number of nuclear VRCs did not affect the number of CA-containing viral cores in the cytoplasm (Fig. 6c). In contrast to a high dose PF74, a low concentration of this drug applied at 4 hpi did not affect VRC localization to NSs, their

number in the nucleus or infectivity, beyond the expected twofold reduction (Fig. 6b–d). This ~50% reduction in infection was due to PF74-mediated block of nuclear import of half of the VRCs that did not get into the nucleus by 4 hpi[13]. The markedly stronger inhibition of infection by Nevi treatment, as compared to a low dose of PF74, is consistent with continued vDNA synthesis in the nucleus.

As stated above, NS-localized VRCs in MDMs were irresponsive to a 30-min treatment with a high dose of PF74 (Fig. 5h) that effectively displaced CPSF6 from NSs (Fig. 4). We reasoned that extensive VRC clustering in comparatively large NSs of MDMs may preclude quick exit of HIV-1 from these compartments. We

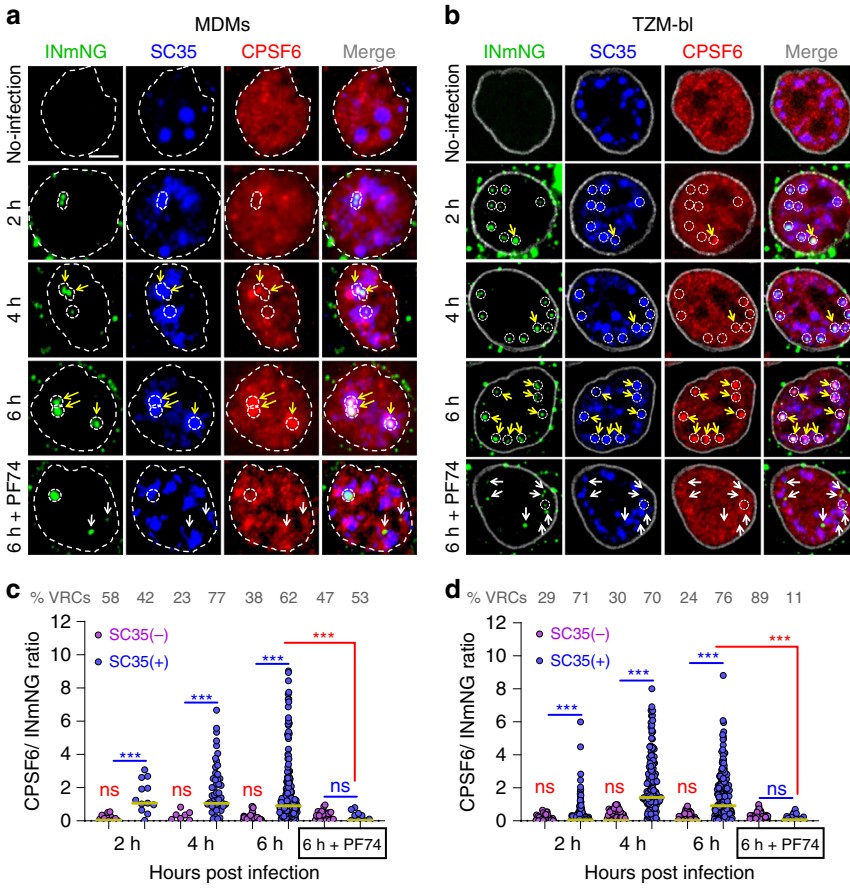

**Fig. 4 HIV-1 VRCs recruit CPSF6 to NSs. a–d** Timecourse of CPSF6 accumulation at NS-localized HIV-1 complexes. TZM-bl cells stably expressing SNAP-Lamin nuclear envelope marker and MDMs (untreated) were infected at MOI of 5 and fixed at indicated time points. Where indicated, PF74 (25 µM) was added to samples 30 min before fixation (denoted 6 h + PF74). Cells were immunostained for NSs (SC35) and endogenous CPSF6. Uninfected cells were used as controls. Association of nuclear INmNG-labeled VRCs with SC35+ compartments in MDMs (**a**) and TZM-bl cells (**b**). White dashed circles and contours show NS-associated IN puncta, yellow arrows point to IN puncta that recruited CPSF6, white arrows in 6 h + PF74 treatment point to the loss of CPSF6 signal from IN puncta. Analysis of the ratio of CPSF6/INmNG fluorescence in nuclear IN puncta residing inside (SC35(+)) and outside (SC35(−)) NSs in MDMs (**c**) and TZM-bl cells (**d**). Note: raw INmNG and CPSF6 fluorescent intensities are shown in Supplementary Fig. 5a–d. Scale bar is 5 µm in (**a**, **b**). Data in (**c**, **d**) are presented as median values (yellow lines) ± SEM at 95% CI. Obtained by analysis of >80 nuclei in each cell type for each time point, from three independent experiments/donors. The number of MDM IN complexes analyzed in (**c**) was 31, 79, 353, and 80 for 2, 4, 6 h, and 6 h + PF74, respectively. In (**d**), >200 IN puncta in TZM-bl cells were analyzed for all time points. Statistical analysis was performed using nonparametric Mann–Whitney rank-sum test (ns$p > 0.05$; ***$p < 0.001$). In (**c**, **d**), analyses for the same sample inside and outside NSs are shown in blue, and the $p$ values for the effect of PF74 treatment at 6 hpi are shown in red. All nuclear IN spots were analyzed without an exception. Source data are provided as a Source Data file.

therefore infected MDMs with INmNG-labeled HIV-1 for 24 h to allow completion of VRC nuclear import and clustering and incubated the cells with PF74 for an additional 5 days. In parallel experiments, we added EdU along with PF74 at 24 hpi to assess the effect on vDNA synthesis. Prolonged exposure to 25 µM PF74 relocated >50% of VRCs from NSs into the nucleoplasm of MDMs (Fig. 7a, b), but did not affect vDNA synthesis in the nucleus (Fig. 7c, d), as measured by the incorporation of EdU into nuclear VRCs, which was blocked by nevirapine. Accordingly, the number of nuclear VRC clusters was not significantly reduced by prolonged PF74 treatment (Fig. 7a). Under these experimental conditions slow/inefficient PF74-mediated exit of VRCs from NSs and their relative stability in the nucleoplasm appear to be associated with a modest effect on infection in these cells (Fig. 7e).

Correlation between the ability of PF74 to quickly relocate VRCs outside of NSs and the observed reduction in the number of nuclear IN puncta and infectivity in TZM-bl cells (Fig. 6b–d) supports a role for NSs in stabilizing HIV-1 VRCs. Moreover, the displacement of CPSF6 and relocation of VRCs from NSs by

PF74 in different cell types (Figs. 4–7 and Supplementary Fig. 5a–f) suggest that CPSF6 is involved not only in HIV-1 transport to NSs, but also in stabilizing the VRC association with these sub-compartments.

**HIV-1 integrates into speckle-associated genomic DNA.** PF74-mediated displacement of HIV-1 from NSs (Fig. 5g, h and Fig. 7a, b) implicated CA-dependent interactions in VRC retention and, possibly, in facilitating integration into nearby loci. To address this possibility, we correlated previously mapped HIV-1 integration sites[19,22,35] with NS-associated genomic domains or SPADs[32]. SPADs are defined as genomic DNA sequences that score in the top 5% of SON-associated (a large Ser/Arg-related protein that, like SC35, localizes to NSs) tyramide signal amplification (TSA)-Seq[33]. We first defined the random integration control (RIC) by mimicking our integration site targeting pipeline methodology in silico, which revealed that 2.8% of randomly chosen genomic fragments mapped to SPADs (dashed line in Fig. 8a–c). Across cell types, HIV-1 highly favored integration into SPADs. While about 30% of HIV-1

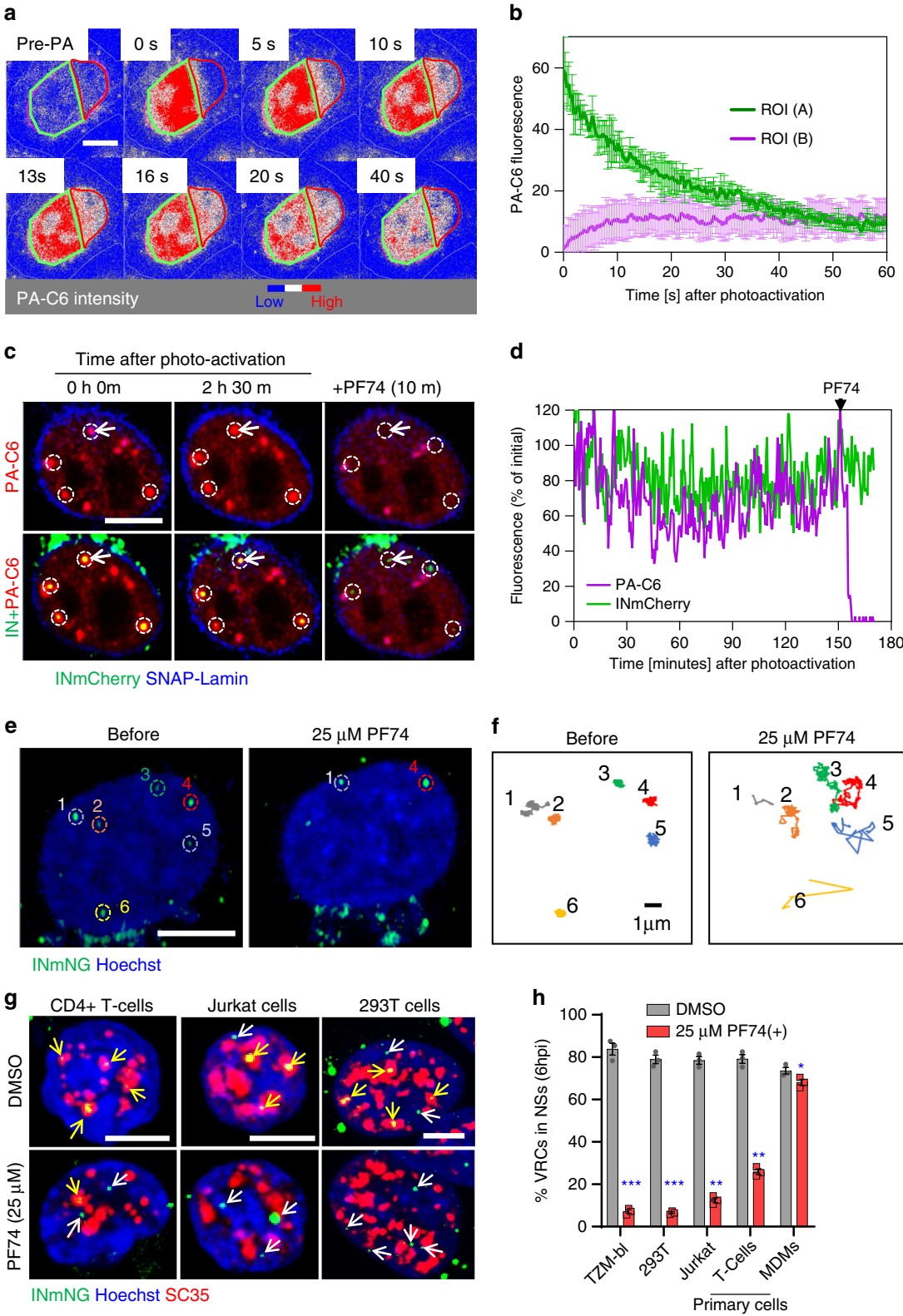

integration occurred within a SPAD (Fig. 8a–c and Supplementary Fig. 6a), the bulk of remaining sites distributed to neighboring chromatin (Fig. 8d–f and Supplementary Fig. 6b).

HIV-1 integration targeting is largely mediated by the interaction of two VRC-associated factors, IN and CA, with lens epithelium-derived growth factor (LEDGF/p75) and CPSF6, respectively (reviewed in Ref. [40]). While integration into SPADs was maintained in HEK293T cells knocked out for LEDGF/p75

(LKO cells), HIV-1 clearly disfavored SPADs for integration in CPFS6 knockout (CKO) cells (Supplementary Fig. 6a, b). Because integration into SPADs remained disfavored in double knockout DKO cells, we concluded that CPSF6 may, in large part, determine SPAD-dependent integration targeting in HEK293T cells (Supplementary Fig. 6a, b).

To address the specificity of CPSF6 in SPAD-tropic targeting, integration sites obtained from CKO cells engineered to express

**Fig. 5 CA-dependent interactions tether VRCs to NSs. a, b** CPSF6 fusion with photoactivatable GFP (PA-C6) transiently expressed in TZM-bl cells is highly mobile in the nucleus. Images (**a**) and quantification (**b**) of redistribution of photoactivated PA-C6 from the illuminated region (green contour, A) into a non-photoactivated region (red contour, B) in a central Z-plane of the nucleus. PA-C6 fluorescence is represented as a heatmap. **c, d** A subset of PA-C6 is tightly associated with nuclear VRCs. Images (**c**) and fluorescent intensity trace (**d**) of INmCherry-labeled nuclear VRCs (dashed white circles) co-trafficking with photoactivated PA-C6 (red) in cells expressing SNAP-Lamin (blue). Imaging started at 4 hpi. PA-C6 was photoactivated at the location of VRCs and live-cell imaging was performed at 20 s/frame for 2.5 h, at which point, addition of 25 μM PF74 displaced PA-C6 from VRCs. The fluorescence intensity trace in (**d**) corresponds to VRC in the top-left corner (white arrow in (**c**)). **e, f** Addition of PF74 results in rapid displacement of VRCs from their location. TZM-bl cells (nuclei stained with Hoechst) were infected with INmNG-labeled HIV-1 for 4 h, and trafficking of nuclear VRCs was imaged at 5 s/frame for 30 min. Images (**e**) and single-particle trajectories (**f**) of nuclear INmNG puncta before and after 25 μM PF74 treatment correspond to VRCs marked by dashed circles in panel (**e**). **g, h** Images and quantification of the fraction of NS-localized VRCs in different cell types untreated or treated with PF74. MDMs, HEK293T, TZM-bl cells, Jurkat cells and primary CD4+ T cells were infected, as described in Fig. 3. Where noted, 25 μM PF74 was added 30 min before fixation at 6 h. Cells were immunostained for NSs (SC35), and the number of IN puncta inside and outside of NSs was determined. Scale bars in (**a**, **c**, **e**, **g**) are 5 μm. Error bars are mean ± SEM. N = 20 nuclei in (**b**) and >120 nuclei in (**h**) from three independent experiments. Significance relative to DMSO control was determined by two-tailed Student's t test (*p = 0.0365; **p < 0.01; ***p < 0.001). Source data are provided as a Source Data file.

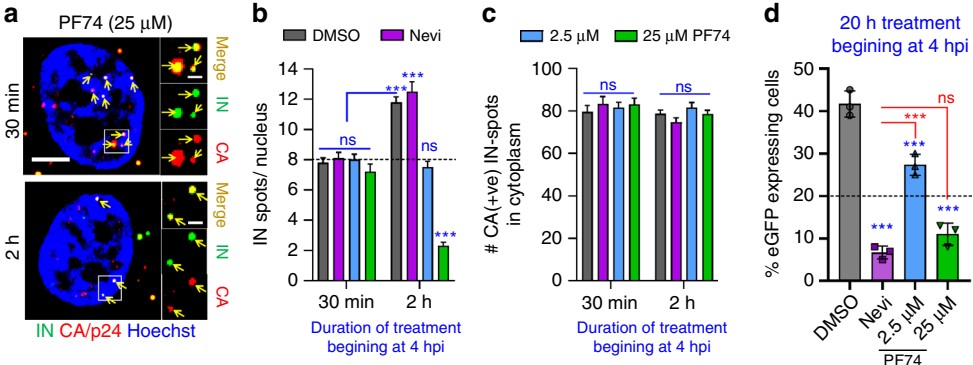

**Fig. 6 NS localization stabilizes VRCs in the nucleus.** TZM-bl cells were infected with VSV-G pseudotyped HIVeGFP labeled with INmNG for 4 h (half-time of nuclear import), at which time DMSO, Nevi (10 μM), or the indicated concentrations of PF74 was added and cells were fixed after 30 min or after 2 h of PF74 application to quantify nuclear IN puncta (**a, b**) and cytosolic CA-positive IN puncta (**c**) or incubated for additional 20 h to measure infection (**d**). **a, b** A high dose of PF74 results in the displacement of IN puncta from NSs and disappearance of IN puncta. **a** Single Z-stack images showing the presence of CA/p24 immuno-labeled (red) IN puncta (green) in the nucleus (blue) of 25 μM PF74 treated cells after 30 min (top panel) or 2 h (bottom panel). Analysis from 30 to 40 cells in each of the three independent experiments showing the average number of IN puncta per nucleus (**b**) or CA-positive IN puncta in the cytoplasm (**c**) is shown. **d** The fraction of eGFP-expressing TZM-bl cells treated with indicated drugs at 4 hpi was determined at 24 h. Dashed lines represent baseline nuclear import upon drug addition (**b**) and 50% of DMSO infection (**d**), respectively. Scale bar in (**a**) is 5 μm and inset in (**a**) is 1 μm. Error bars in panels (**b–d**) are mean values ± SEM. Data in (**b, c**) are from >80 nuclei/cells containing >1500 IN complexes and three independent experiments (**a–d**). The statistical significance in (**b–d**) was determined by two-tailed Student's t test (ns, p > 0.05; ***p < 0.001). Results of statistical analyses in (**d**) vs. DMSO control is shown in blue vs. Nevi that is shown in red. Statistical analysis in (**b**) with respect to a 30 min treatment with DMSO is shown in blue. Source data are provided as a Source Data file.

WT vs. mutant F284A CPSF6, which is defective for CA binding, were analyzed. While WT CPSF6 in large part restored integration targeting of SPAD regions, F284A CPSF6 failed to do so (Fig. 8a, d). Accordingly, CPSF6 interaction-defective N74D[9] and A77V[35] CA mutant viruses were defective for SPAD integration targeting in HEK293T cells, as well as in primary MDMs and CD4+ T cells (Fig. 8b, c, e, f and Supplementary Fig. 6a, b). In contrast to MDMs infected with replication-competent HIV-1 (Bal in Fig. 8b, e), we do note that SPAD integration targeting by the A77V mutant remained significantly enriched compared to the RIC in primary T cells (Fig. 8c, f).

Lucic et al.[41] recently reported that there could be two processes governing HIV-1 integration site selection. One, attracting the virus to active genes and a second, facilitating integration into recurrent integration genes (RIGs), which are genes that are repeatedly targeted for integration independent of cell type[42]. Nuclear architecture, including elements, such as the nuclear periphery[42] and super enhancers (SEs)[41], has been proposed to mediate RIG-tropic integration. Because SEs are known to associate with NSs[32,43,44], we next correlated bulk integration sites for SE proximity. Consistent with Lucic et al.[41], HIV-1 integration into SE regions in WT HEK293T cells was not

enriched compared to the RIC. While this phenotype was largely preserved in CKO cells, HIV-1 gained a marginal preference for SEs in LKO cells (Supplementary Fig. 6c).

Confirming that SEs are not a main driver of bulk HIV-1 integration targeting, we next analyzed RIGs for proximity to SPADs and SEs. We curated (Supplementary Table 1) 46 CPSF6 (+) RIGs from multiple cell types (HEK293T, HOS, MDM, and CD4+ T cells) and 30 CPSF6(−) RIGs from these same cells when infection occurred in the absence of the CA–CPSF6 interaction (i.e., N74D or A77V CA mutant viruses, or WT CA infection of CKO cells). The bulk of CPSF6(+) RIGs, 43 of 46, rather harbored a SPAD within the gene, with a mere 3.1 kb representing the mean distance (range 0–0.11 Mb) of all RIGs from SPADs. In rather stark contrast, none of the CPSF6(−) RIGs associated with SPADs, with a mean distance of 19.9 Mb to the nearest SPAD (range 3.2–57.6 Mb), more than three orders of magnitude removed from SPADs than normal (Fig. 8g). Consistently, proximity of CPSF6(−) RIGs to SEs (mean distance of 2.7 Mb; range 0.004–11.8 Mb) also diminished when the virus could not engage CPSF6 (mean distance 8.0 Mb; range 0–71.0 Mb), though this approximate threefold difference in mean RIG distance to target was less pronounced than the difference noted

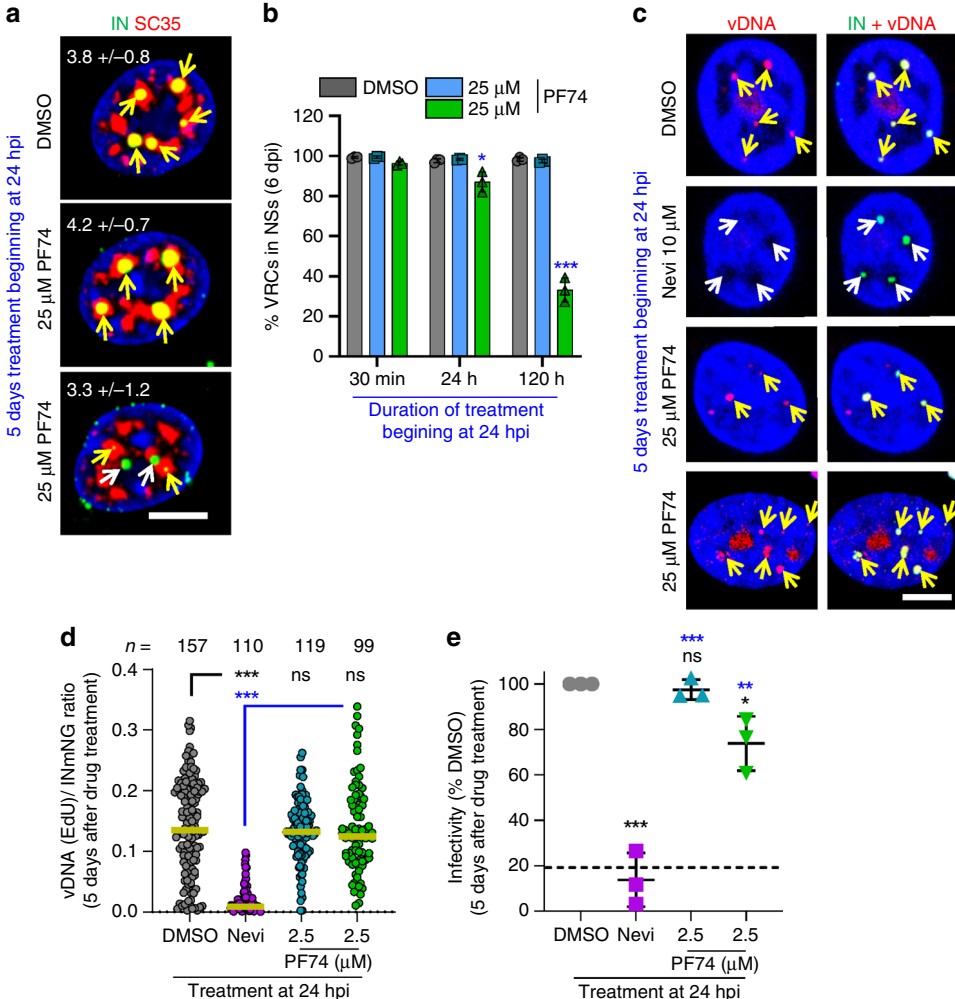

**Fig. 7 PF74-resistant nuclear VRC clusters in MDMs continue to synthesize vDNA.** Treatment with 25 μM PF74 promotes slow exit of VRCs from NSs of MDMs, without affecting vDNA synthesis. MDMs were infected with INmNG-labeled HIV-1 (**a–d**) for 24 h, at which time PF74, DMSO, or Nevi was added without (**a**, **b**, **e**) or with 5 μM EdU (**c**, **d**), and incubated for different time, as indicated. Cells were fixed and stained for NSs (SC35) (**a**, **b**) or vDNA (**c**, **d**). **a**, **b** Images and quantification of the fraction of nuclear VRCs colocalized with SC35 compartments; **c**, **d** images and EdU/vDNA staining of nuclear IN puncta after treatment with DMSO and PF74 at 24 hpi. Fluorescence intensities of EdU and INmNG of nuclear VRCs in panel (**d**) were measured and the ratio of EdU/vDNA to INmNG signals was plotted. **e** HIV-1 infection of MDMs becomes resistant to PF74 treatment after 24 h. MDMs infected with VSV-G pseudotyped pNL4.3 R-E-Luc were treated with indicated drugs beginning at 24 hpi. Cells were cultured for additional 5 days in the presence of drugs; luciferase activity was measured in triplicate and normalized to DMSO control. Each data point in (**e**) represents infectivity from an independent donor, dashed black line denotes the fraction of completion of vDNA synthesis at the time of drug addition (24 h). Scale bars in (**a**, **c**) are 5 μm. The average number of nuclear VRCs (±SD) detected in >60 nuclei from three experiments is overlaid on images in (**a**). Mean values ± SEM are shown in panels (**b**, **e**). Yellow line in (**d**) shows median values at 95% CI from three independent experiments/donors for >80 nuclei; the total number of nuclear IN puncta analyzed is shown. Significance in (**b**) and (**e**) (black—relative to DMSO, blue—nevirapine) was determined by two-tailed Student's t test. p values in (**d**) were determined by a nonparametric Mann–Whitney rank-sum test. *p = 0.0228 in (**b**), p = 0.0138 in (**e**); ***p < 0.001 in (**b**, **d**, **e**). Images in (**a**, **c**) are representative of >120 nuclei, from three independent experiments/donors. Source data are provided as a Source Data file.

for SPADs (Fig. 8g, h and Supplementary Table 1). Collectively, our results support the notion that CPSF6-mediated transport of HIV-1 VRCs to NSs determines bulk integration sites and the 3D compartmentalization of the genome for recurrent integration.

**Transcription factors are enriched at nuclear HIV-1 clusters.** We next asked whether transcription of viral genes can occur at sites of nuclear VRC clusters by using multiplex immunofluorescent cell-based detection of DNA, RNA, and protein, which is based on branched DNA in situ hybridization[45]. While treatment with 2.5 μM PF74 inhibited VRC nuclear import, ~4–6 HIV-1 clusters persisted in Nevi- or RAL-treated cell nuclei (Fig. 8i, j). Infected MDMs contained a markedly greater number

of vRNA spots per nucleus compared to uninfected cells or cells infected in the presence of PF74 or Nevi (Fig. 8l). By contrast, a very modest increase in the number of vRNA puncta compared to background was detected in RAL-treated cells (Fig. 8l). Immunostaining of infected MDMs for components of the P-TEFb complex, which is recruited to the transcribing viral genome (reviewed in Ref. [46]), revealed that Cyclin-dependent kinase 9 phosphorylated at Ser175 (CDK9/pS175) accumulates at nuclear HIV-1 clusters that also stained positive for vRNA, vDNA, and IN (~18%) (Fig. 8i). The co-appearance of P-TEFb with vRNA and vDNA signals at the sites of HIV-1 clustering is consistent with productive integration at these loci, as also reported by others[18].

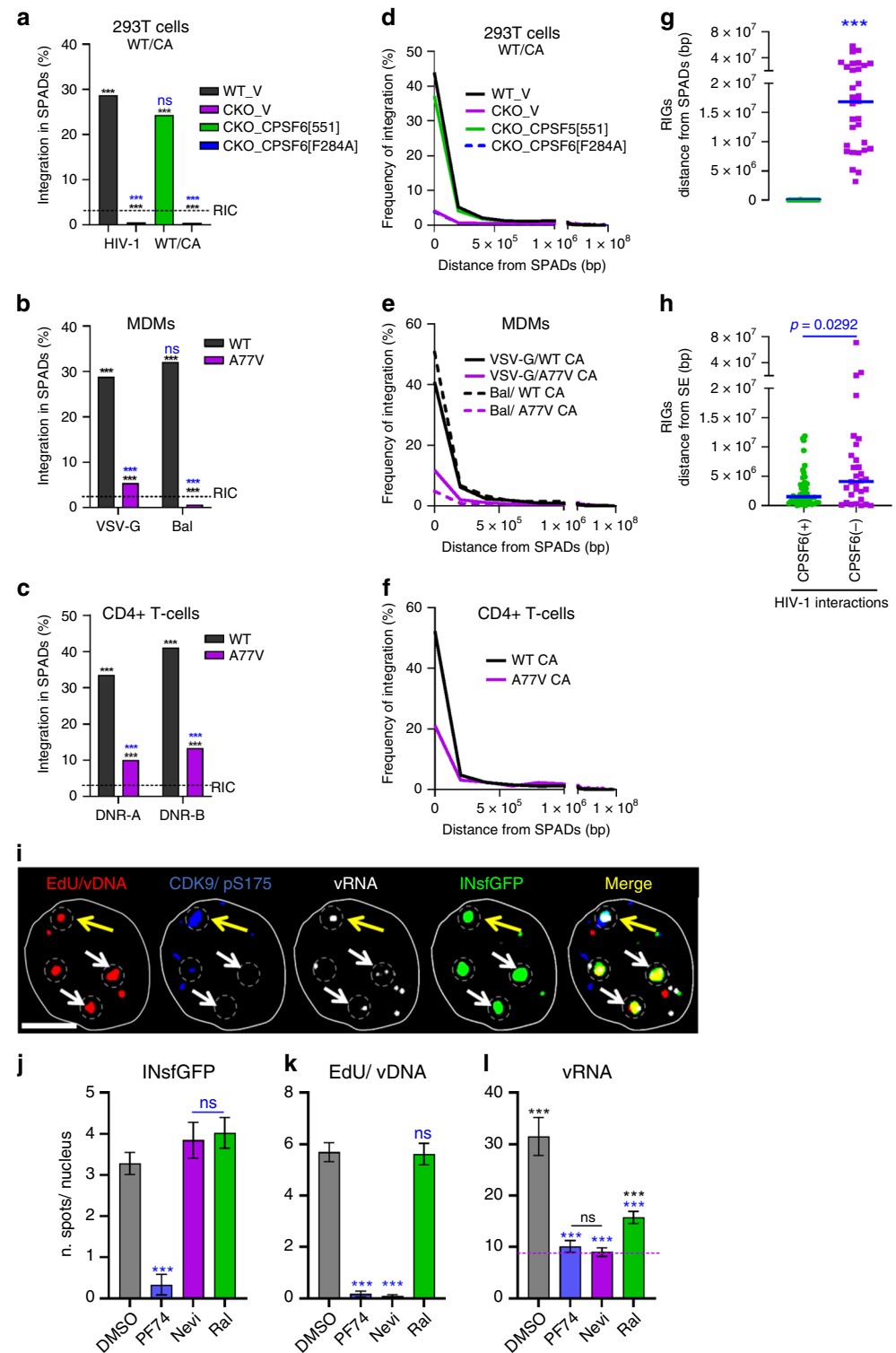

## Discussion

Quantitative imaging of HIV-1 infection demonstrated the formation of large IN-containing complexes in MDM cell nuclei. Further investigation of the intranuclear transport of VRCs revealed their accumulation in NSs, which are the preferred destinations of HIV-1 in MDMs and other cell types. HIV-1 transport to NSs is mediated by the CA–CPSF6 interaction but independent of reverse transcription. We also provide evidence that, at least in TZM-bl cells, the localization of VRCs in NSs stabilizes these complexes and facilitates infection. Importantly,

HIV-1 targeting to NSs explains its integration site preference, as we uncover a marked preference for integration into SPAD regions of the human genome. Whereas HIV-1 association with NSs has been verbally communicated by Vineet KewalRamani (communication in Ref. [19]), here we report the prevalence of this trafficking pathway in multiple cell types and its importance for preferential HIV-1 integration targeting into SPADs.

CA plays multiple roles during the early stages of HIV-1 infection, including evasion of cellular innate defenses, mediation of virus nuclear import and, through interaction with CPSF6,

**Fig. 8 HIV-1 integration favors NS-associated genomic loci. a, d** Analysis of HIV-1 integration preferences into SPAD regions in WT and CKO HEK293T cells transduced with empty expression vector (WT_V and CKO_V, respectively) or CKO cells transduced with vector expressing WT or F284A CPSF6 protein. **b, e** MDMs infected with single-round VSV-G pseudotyped or replication-competent (Bal) virus. **c, f** Primary CD4+ T cells derived from two blood donors A and B. **a–c** Percentage of HIV-1 integration sites within SPAD regions. **d–f** Frequency distribution of all integration sites as a function of their distance from closest SPAD, bin sizes are 200 kb. **g, h** CPSF6-dependent integration targeting of SPADs. Distances of RIGs from SPADs (**g**) vs. SEs (**h**). In (**a–c**), p values relative to matched WT conditions (blue asterisks) and to RIC (black asterisks) were calculated by Fisher's exact test. A nonparametric Mann–Whitney rank-sum test was used in (**g, h**; ns$p > 0.05$; ***$p < 0.0001$). **i–l** Vpx(+) MDMs were infected with INsfGFP labeled VSV-G pseudotyped HIV-1 at MOI of 2 in the presence of EdU (5 μM). Infections were carried out in the presence of PF74 (2.5 μM), Nevi (10 μM), or RAL (10 μM) fixed at 72 hpi, and stained for CDK9/pS175, EdU/vDNA, and transcribed vRNA. **i** Images of an MDM nucleus showing INsfGFP (green), EdU (red), CDK9/pS175 (blue), and vRNA (white) spots. IN clusters are marked by dashed circles. White arrows point to IN clusters colocalized with EdU and vRNA spots; a yellow arrow shows colocalization of a single IN cluster with all three markers, including CDK9/pS175. Quantification of the number of nuclear IN (**j**), nuclear EdU (**k**), and vRNA puncta (**l**) is shown. Scale bar in (**i**) is 5 μm. Error bars in (**j–l**) are mean values ± SEM for >90 nuclei from three donors. The p values relative to DMSO are shown in blue (**j–l**) and relative to background (BG) RNA spots detected by RNAscope in noninfected MDMs are in black (**l**) (dashed blue line) was determined by two-tailed Student's t test (ns$p > 0.05$; **$p < 0.01$; *** $p < 0.001$). Source data are provided as a Source Data file.

transport of VRCs away from the nuclear periphery to sites of vDNA integration (reviewed in Refs. [40,47]). Here, we provide evidence for a marked loss of the HIV-1 CA marker CDR (uncoating) during nuclear import in MDMs and demonstrate the critical role of residual nuclear VRC-associated CA in trafficking and retention of these complexes in NSs. The observed robust CA signal associated with nuclear VRCs in MDMs has been previously interpreted as evidence against HIV-1 uncoating prior to nuclear import in these cells[14,15,28,48]. However, our results (Fig. 1 and Supplementary Fig. 2) demonstrate that, under similar experimental conditions, the nuclear CA signal in MDMs originates primarily from clustering of multiple VRCs and not from intact viral cores.

PF74 competes with CPSF6 and other essential host factors for binding to a pocket at the NTD-CTD interface in the hexameric CA structure[49–52]. The resistance of nuclear VRC-CPSF6 complexes to a low dose of PF74 is consistent with the high-affinity binding between CA and CPSF6 in NSs. The ability of a high dose of PF74 to displace CPSF6 from nuclear VRCs and induce relocation of VRCs from NSs suggests that CPSF6 and/or other host-factors sharing the same CA binding pocket are involved in tethering HIV-1 complexes to speckles. We also provide evidence that, at least in TZM-bl cells, the relocation of VRCs outside of NSs diminishes the stability of VRCs and reduces infectivity. It is possible, however, that a high concentration of PF74 could have off-target effects, so additional studies of the VRC stability outside speckles are warranted.

Interestingly, in contrast to a recent report[53], we did not observe reduction in CA signal associated with nuclear VRCs upon PF74 treatment of infected HeLa cells (Fig. 6c and Supplementary Fig. 5g, h). In contrast, we observed a selective enhancement of nuclear VRC-associated CA signal by PF74 compared to cytoplasmic complexes (Supplementary Fig. 5g, h), highlighting a fundamental difference in the structure and/or composition of these complexes. We surmise that the reported nuclear CA-destabilizing effect of PF74[53] may be related to the use eGFP-CA fusion protein for HIV-1 core labeling. Moreover, in MDMs, where multiple VRCs form stable clusters (Supplementary Fig. 3), a high dose of PF74 added after cluster formation had no effect on VRC cluster-associated vDNA synthesis (Fig. 7c, d). Because a high dose of PF74 has been reported to block reverse transcription when added early during infection of HeLa cells[37], by virtue of altering virus core-stability, this somewhat surprising result suggests that PF74 may not affect reverse transcription outside the context of an intact conical core.

The mechanism of stable HIV-1 cluster formation in MDM nuclei and its role in infection are unknown. Given that the A77V/CA mutant virus, which is able to replicate in MDMs, does not reach NSs or form clusters (Fig. 2c–f and Supplementary

Fig. 4c, e), HIV-1 transport to and accumulation of VRCs in NSs appear largely dispensable for HIV-1 infection. However, given the high propensity for HIV-1 to integrate into SPADs (Fig. 8), the localization of VRCs in NSs is functionally conserved. Accordingly, in growth competition experiments, WT HIV-1 that can bind CPSF6 consistently outpaces A77V mutant virus that is defective for CPSF6 binding[35].

RIG analyses initially suggested that the nuclear periphery provides an architectural basis for HIV-1 integration targeting, as some RIGs mapped to the peripheral region of the nucleus independent of virus infection[42]. More recently, Lucic et al. showed significant association of RIGs with regions of the genome enriched in SEs[41]. Notably, analysis of bulk integration sites failed to reveal significant association with SE regions (Ref. [41] and Supplementary Fig. 8c). By contrast, our analysis of SPAD regions of chromosomes, which are proximal to NSs[33], revealed strong positive correlation for bulk HIV-1 integration. Moreover, HIV-1 integration targeting of SPAD loci strictly depended on the CA–CPSF6 interaction.

To assess the relative importance of SPAD vs. SE-associated genomic regions in integration targeting, relative positions of CPSF6(+) and CPSF6(−) RIGs were determined. Consistent with the recent report, CPSF6(+) RIGs clustered nearby SE regions independent of cell type (Fig. 8h) and genes distal from SEs were repeatedly targeted for integration in the absence of CA–CPSF6 (Fig. 8h). These observations are consistent with our prior report that CPSF6(−) RIGs associate with LADs, which are heterochromatic regions at the nuclear periphery[19]. Though CPSF6(+) RIGs are closer to SEs than are CPSF6(−) RIGs, this difference pales in comparison to the CPSF6-dependent difference observed for RIG association with SPADs (compare Fig. 8h with Fig. 8g). Moreover, the vast majority of CPSF6(+) RIGs, including NPLOC4 and NFATC3 that preferentially map to the nuclear periphery in the absence of HIV-1 infection[19,42], harbor a SPAD region (Fig. 8g and Supplementary Table 1).

High-resolution mapping of genome-wide chromatin interactions using Hi-C has revealed that the genome spatially segregates into distinct compartments. Transcriptionally active regions cluster into A1 and A2 sub-compartments[41,54]. Although the coverage of euchromatin marks and the transcriptional activity of A1 are only slightly enriched as compared to A2, HIV-1 intrinsically favors integration into A1 sub-compartment chromatin[41]. Consistent with our findings, SON TSA-Seq mapping correlated the A1 sub-compartment with NSs[33]. It seems evident to us that CPSF6 trafficking of HIV-1 VRCs to NSs determines the intrinsic affinity of HIV-1 to integrate in A1 sub-compartment chromatin. We accordingly conjecture that NSs, and not the nuclear periphery as first espoused, provide the architectural basis for HIV-1 integration targeting within the cell nucleus.

## Methods

**Plasmids.** The CypA-DsRed (CDR), Vpr-IN-superfolderGFP (INsfGFP), and Vpr-INmNeonGreen (INmNG) expressing plasmids have been described previously[13,55,56]. Plasmids pR9ΔEnv[8] and pNL4.3 R-E-Luc[57] were previously described. The pMD2.G vector expressing VSV-G glycoprotein was a gift from Dr. J. Young (The Salk Institute for Biological Studies, La Jolla, CA) and the psPAX2 vectors obtained from Addgene (Cat# 12260) was a gift from Didier Trono. The pHIVeGFP (NL4.3 R-E-eGFP) and pBru-A77VeGFP plasmids were a gift from Dr. Christopher Aiken (Vanderbilt University) and Dr. Masahiro Yamashita (Aaron Diamond AIDS Research Center), respectively. The NL4.3-Nef-HA plasmid was provided by Dr. Massimo Pizzato (University of Trento). The pLenti.SNAP-LaminB1-10 construct was made by swapping the EBFP2-encoding fragment from EBFP2-LaminB1-10 (Addgene #55244), using AgeI and XhoI, and cloning into pLenti.EBFP2-LaminB1-10 plasmid described previously (Francis and Melikyan[13,28]). Plasmids encoding for SIV-Gag/Gag-Pol Vpx(+) and SIV Vpx(−) was a gift from Dr. Nathaniel Landau[58]. The PAeGFP was amplified from pACAGW-H2B-PAGFP-AAV plasmid (Addgene, #33000) and cloned into pcDNA3_3xFLAG-MBP-BamH-CPSF6(FL)-HindIII-His plasmid (a kind gift from Jinwoo Ahn, University of Pittsburgh) by replacing the maltose binding protein (MBP) at the amino terminus of CPSF6 using enzymes AgeI and BamHI.

**Cell lines and reagents.** HEK293T/17 cells (from ATCC, Manassas, VA) and HeLa-derived TZM-bl cells (from NIH AIDS Reference and Reagent Program) were grown in high-glucose Dulbecco's Modified Eagle Medium (DMEM, Mediatech, Manassas VA) supplemented with 10% heat-inactivated fetal bovine serum (FBS, Cat# S11150H, Atlanta Biologicals, Flowery Branch, GA) and 100 U/ml penicillin–streptomycin (Gemini Bio-Products, Sacramento, CA). The growth medium for HEK293T/17 was supplemented with 0.5 mg/ml G418 sulfate (Mediatech, Manassas VA). Jurkat T cells were a kind gift from Drs. D. Braaten and J. Luban[59] and were maintained in RPMI complete media (Mediatech, Manassas VA) supplemented with 10% FBS and 100 U/ml penicillin–streptomycin. SNAP-LaminB1-10 expressing TZM-bl cells were generated using a lentiviral vector pLenti.SNAP-LaminB1-10 encoding for LaminB1. Cells were subjected to limited dilution and selection of clones expressing optimal levels of SNAP-Lamin. All cell lines were cultured in a 37 °C incubator supplemented with 5% CO$_2$, and were passaged at regular intervals.

Cyclosporin A was obtained from Calbiochem (Burlington, MA), dissolved in DMSO at 50 mM and stored in aliquots at −20 °C. PF74 (#PF-3450074), Aphidicolin (#A0781), Nocodazole (M1404), Actinomycin D (# A9415), 1,6-hexanediol (#240117), 5,6-dichlorobenzimidazole 1-β-D-ribofuranoside (#D1916), Coumermycin (C-A1) (#C9270), and mouse anti-tubulin antibody (#T6074) were purchased from Sigma-Aldrich. Alpha-amanitin was purchased from Tocris Bioscience (#4025, Minneapolis, MN). TNFα was from Genscript, Piscataway, NJ, USA. Bright-Glo luciferase assay kit was from Promega (Madison, WI). Puromycin was obtained from InvivoGen. Antibodies to LaminB1 (#ab16048), CypA (#ab3563), NS marker SC35 (mouse ab11826, rabbit ab204916), SAMHD1 (mouse polyclonal antibody abcam, cat# ab67820), and donkey anti-rabbit AF405 antibody (#ab175651) were purchased from Abcam (San Francisco, CA). Rabbit anti-SON (polyclonal IgG) was from AtlasAntibodies (#HPA031755). Cy5-conjugated anti-mouse antibody was made from SouthernBiotech (Birmingham, AL). CPSF6 (Rabbit PA5-41830; Invitrogen), CyclinT1 (mouse sc-271348; Santa Cruz Biotechnology) mouse monoclonal primary antibody against CDK9/ pS175 was a kind gift from Dr. Jonathan Karn (Case Western University). Trypan blue was purchased from Sigma-Aldrich (cat# T8154). CellTiter-Blue Cell Viability Assay Promega (cat# G8080). The SNAP-InCellStarRed and SNAP-Cell® 647-SiR dyes were purchased from New England Biolabs (NEB, #S9102S). Phosphate buffered saline containing Mg$^{2+}$/Ca$^{2+}$ (dPBS) and Mg$^{2+}$/Ca$^{2+}$-free (PBS) were purchased from Corning (MediaTech, Manassas, VA). EdU (#A10044), Click-iT®EdU Imaging kits (#C10338 and #C10340), and Hoecsht33442 (#62249) were from ThermoScientific.

The following reagents were obtained from the NIH AIDS Reference and Reagent Program, Division of AIDS, NIAID, NIH: pNL4-3.Luc.R-E- from Dr. Nathaniel Landau[57,60], TZM-bl cells expressing CD4, CXCR4, and CCR5 from Drs. J.C. Kappes and X. Wu[61]; anti-p24 antibody AG3.0 donated by Dr. J. Alan[62]; RT inhibitor Nevirapine and IN inhibitor Raltegravir (Merck & Company).

**Isolation and differentiation of MDMs.** De-identified human blood samples for preparation of MDMs and CD4+ T cells were obtained from volunteer donors after informed consent (approved by Emory IRB000045690 "Phlebotomy of Healthy Adults for Research in Infectious Diseases and Immunology"). Human peripheral blood mononuclear cells (PBMCs) were isolated from fresh heparinized blood by Ficoll-Hypaque gradient centrifugation. PBMCs from buffy coats were pooled and extensively washed to remove platelets. Monocytes were isolated by magnetic labeling using Monocyte Isolation Kit II (Miltenyi Biotec Inc) according to the manufacturer's protocol. Enriched monocytes were adhered to poly-D-lysine coated dishes. Briefly, $1 \times 10^6$ cells were plated in six-well plates from Corning (# 3904, MediaTech, Manassas, VA) and 35 mm MatTek dishes (#P35GCOL-1.5-10.C; MatTek Corporation, Ashland, MA) for live-cell imaging. For fixed cell imaging studies $5 \times 10^4$ cells were plated in eight-well chamber coverslips (MatTek, Corp). Monocytes were maintained in RPMI-1640 supplemented with 10% FBS, 100 µg/ml streptomycin, 100 U/ml penicillin, 2 mM glutamine, and 5 ng/ml GM-CSF (Gemini Bio-Products 300124P020G70K). Monocyte cultures were maintained in cytokine-supplemented medium for 7 days to facilitate terminal differentiation into MDMs. Fresh medium was supplied every 2–3 days, and all experiments were performed on days 1–21 after GM-CSF removal. For infectivity assays, $1 \times 10^6$ terminally differentiated MDMs from six-well plates were collected by Trypsin/EDTA treatment for 15 min and plated in a 96-well plate from Corning (Corning, Kennebunk ME) at $1 \times 10^4$ cells/well.

**Depletion of SAMHD1 in MDMs.** Monocytes ($5 \times 10^6$ cells) in suspension were treated with SIV VLPs Vpx(+) or (Vpx−) containing 10 RT units (RTU) for 2 h at 37 °C, 5% CO$_2$. The cells were then plated in GM-CSF containing RPMI medium. Where noted, Vpx(+) treated monocytes were immediately transduced with lentiviral vectors encoding the SNAP-LaminB1-10 nuclear envelope marker and cultured for 24 h prior to differentiating in GM-CSF containing media for additional 7 days. Lentiviral transduction was enhanced by a 30 min spin at $1500 \times g$ at 16 °C and additional incubation for 24 h. Alternatively, Vpx(+) treatment was performed on differentiated MDMs on the day of GM-CSF removal by briefly spinning SIV VLPs (5 RTU) for 30 min at $1500 \times g$, 16 °C followed by a 2 h incubation in a 37 °C CO$_2$ incubator. In all cases, the SIV VLPs were washed off and MDMs were further cultured in fresh RPMI supplemented with 10% FBS. The stable depletion of SAMHD1 by Vpx(+) treatment was verified at 1, 7, and 14 days after GM-CSF removal by fluorescence microscopy following immune-staining using primary anti-SAMHD1 mouse polyclonal antibody (Abcam, cat# ab67820) and Cy5-conjugated anti-mouse antibody from SouthernBiotech (Birmingham, AL). Nearly all cells treated with Vpx(+) showed efficient SAMHD1 depletion. When treating cells with high-dose 25 µM PF74 for prolonged period of time, cell-viability measurements were performed based on Trypan blue staining of dead cells or using the CellTiter-Blue Cell Viability Assay kit, as per the manufacturer's protocol and as described in Supplementary Fig. 7.

**Isolation and activation of CD4+ T cells.** CD4+ T cells were isolated from PBMCs by magnetic labeling using CD4+ T-cell Isolation Kit (130-096-533; Miltenyi Biotec Inc) according to the manufacturer's protocol. Isolated naïve CD4+ T-cell cultures were activated using 10 ng/ml IL-2 and 2.5 µg/ml phytohaemagglutinin cytokine-supplemented media for 3 days, after which the activated CD4+ T cells were pelleted and washed twice in dPBS and incubated in fresh RPMI supplemented with 10% FBS, 100 µg/ml streptomycin, 100 U/ml penicillin, 2 mM glutamine. Resulting cells were used for experiments for up to 10 days. Infection of CD4+ T cells and Jurkat cells was done in suspension by incubating with viral supernatant for 2 h, followed by a brief wash in dPBS and further incubation in RPMI until the time points indicated in the experiments.

**Pseudovirus production and characterization.** Fluorescently labeled pseudoviruses were produced and characterized, as described in Ref. [55] and as follows. HEK293T/17 cells grown in six-well culture plates were transfected with the following plasmids: HIV-1 pR9ΔEnv (2 µg), VSV-G (0.2 µg), Vpr-INmNG (0.5 µg) and, where indicated, CypA-DsRed (0.5 µg), using the JetPrime Transfection reagent (VWR, Radnor, PA). The eGFP reporter-encoding fluorescent HIV-1 pseudoviruses were produced by co-transfecting VSV-G (0.2 µg), pHIVeGFP (2 µg), Nef-HA (0.4 µg) and, where indicated, Vpr-INmNG (0.5 µg) and/or CypA-DsRed (0.5 µg). The A77VeGFP CA mutant virus was generated using pBru-CA A77VeGFP deltaEnv plasmid instead of pHIVeGFP plasmid. Where indicated, red fluorescent protein-labeled viruses were produced by replacing Vpr-INmNG with Vpr-INmCherry plasmid. For generating SNAP-Lamin expressing lentivirus, pLenti.SNAP-LaminB1-10 (2 µg) was co-transfected with the Gag-Pol expressing psPAX2 (1 µg) and VSV-G (0.2 µg) vectors. SIV VLPs encoding or not encoding Vpx (Vpx(+) or (Vpx−)) were generated by transfecting respective Gag-Pol plasmids[58] (2 µg) and VSV-G (0.2 µg).

Six hours after transfection, the medium was replaced with 2 ml of fresh DMEM/10% FBS without phenol red, and the samples were incubated for additional 36 h at 37 °C, 5% CO$_2$. Viral supernatant was collected, filtered through a 0.45 µm filter and quantified for p24 content using AlphaLISA immunoassay kit (PerkinElmer, Waltham, MA) or RT activity (RTU) measured using the PERT protocol[13,63]. For live-cell imaging, fluorescent viruses were purified through a 20% sucrose cushion or concentrated 10× using LentiX concentrator (Clontech Laboratories, Inc Mountainview, CA). Concentrated viruses were resuspended in FluoroBrite (GIBCO) or RPMI-1640 medium containing 10% FBS, aliquoted and stored at −80 °C. MOI was determined in TZM-bl cells by examining the %eGFP-expressing cells after 48 h of infection with VSV-G pseudotyped HIVeGFP virus.

**Single-cycle infection assay.** Terminally differentiated MDMs were plated onto a 96-well plate from Corning (Corning, Kennebunk ME) at $10^4$ cells/well. After 48 h, triplicate wells were infected with VSV-G pseudotyped pNL4.3 R-E-Luc (0.2 ng p24). Virus binding to cells was enhanced by a 30 min spin at $1500 \times g$, 16 °C. Cells were cultured at 37 °C for 120 h, lysed, and luciferase activity measured using the bright glow luciferase substrate (Promega). Where indicated, raltegravir (10 µM), nevirapine (10 µM), or PF74 (2.5 µM or 25 µM) were used at indicated times after infection. For image-based quantification of infected cells, $4 \times 10^4$ MDMs were differentiated in an eight-well chamber slide with GM-CSF for 7 days. After

GM-CSF removal, cells were infected with a serially diluted VSV-G pseudotyped HIVeGFP virus and incubated for up to 5 days. MDMs were fixed and the % of infected cells was determined by counting the number of eGFP reporter expressing cells from four random fields of view and normalizing to the total number of cell nuclei determined by Hoechst staining.

**Fixed cell imaging and immunofluorescence assay**. For fixed cell imaging, MDMs, TZM-bl, or HEK293T/17 cells ($5 \times 10^4$ each) or $1 \times 10^6$ Jurkat or primary CD4+ T cells were infected at MOI 0.5–5 with fluorescent viruses using spinoculation, as above, and washed prior to incubation at 37 °C in 5% CO$_2$. Note: infection of activated CD4+ T cells was between 5 and 20% in three different experiments. Where indicated, EdU (5 μM) was added at respective time points of MDM infection and maintained until cell fixation, which was followed by DNA detection using the manufacturer's protocol. EdU incorporation into vDNA during reverse transcription is readily detected by click-labeling with a fluorescently tagged alkyne[14,15,18,64]. The non-cycling nature of terminally differentiated MDMs makes these cells suitable for analyzing EdU incorporation into vDNA[14,15,18]. Drugs were added at indicated concentrations immediately after virus binding. Cells were fixed with 4% PFA (Electron Microscopy Sciences, #1570-S) for 7 min at indicated time points. Suspension Jurkat and CD4+ T cells were pelleted and fixed on eight-well chamber poly-D-lysine coverglass for 30 min at 4 °C. Fixed cells were permeabilized with 0.1% TX-100 for 5 min at room temperature, washed and blocked in 3% BSA with 0.1% Tween-20 in dPBS. Primary antibody AG3.0 against CA/p24 (diluted 1:100), anti-LaminB1 antibody (1:1000), anti-CPSF6 (1:200), anti-SC35 (1:200), anti-SON (1:200), anti-CycT1 (1:200), anti-CDK9-pS175 (1:200) diluted in the blocking solution (3% BSA 0.1% Tween-20) was allowed to bind for 1 h at room temperature or overnight at 4 °C. Cells were washed five times with 0.1% Tween-20 in PBS and incubated with secondary goat anti-mouse Cy5 antibodies (1:1000), washed 5× and incubated with goat anti-rabbit-AlexaFluor405 or AF555 antibodies (1:1000), each for 1 h at room temperature.

For RNAscope experiments, permeabilized cells were treated with 1:15 diluted Protease 3 (Advanced Cell Diagnostics, #322337) in PBS and incubated in a humidified HybEZ oven at 40 °C for 15 min. Samples were washed in PBS twice and fixed again with 4% PFA for 5 min followed with three washes in PBS. HIV-nongagpol-C3 probe (#317711-C3, Advanced Cell Diagnostics) was added to the samples and incubated in humidified HybEZ oven at 40 °C for 2 h. Two consecutive wash steps were performed with agitation at room temperature for 2 min using the proprietary wash buffer in every wash step after this point, and all incubations were performed in humidified HybEZ oven at 40 °C. The probes were visualized by hybridizing with amplifiers (Amp) and fluorescent label. Amp 1-FL was added to the coverslip for 30 min, followed by Amp 2-FL hybridization for 15 min. Amp 3-FL was then added for 30 min, followed by Amp 4A-FL hybridization for 15 min, labeling the C3 probe with Atto647.

**Live-cell imaging of HIV-1 uncoating and nuclear entry**. HIV-1 uncoating and nuclear import in live cells were visualized as previously described[13]. In brief, VSV-G pseudotyped pR9ΔEnv or pHIVeGFP particles co-labeled with INmNG and CypA-DsRed were bound to 5·10^5 differentiated MDMs by spinoculation at $1500 \times g$ for 30 min, 16 °C. Nuclear import was monitored between 0 and 40 hpi after infecting cells with 500 pg of p24 (MOI 0.5). Where mentioned, MDMs treated with Vpx(+) and expressing the SNAP-Lamin nuclear envelop marker were labeled for 30 min with SNAP-Cell® 647-SiR dyes (NEB, #S9102S) prior to virus binding. Alternatively, MDM nuclei were stained for 10 min with 2 μg/ml Hoechst-33342. Next, viruses were bound to cells by spinoculation. Following spinoculation, the cells were washed twice, and virus entry was synchronously initiated by adding pre-warmed complete RPMI-1640 medium (Gibco) to samples mounted on a temperature- and CO$_2$-controlled microscope stage. Where indicated, live-cell imaging was performed at later time points, as indicated in figure legends, in TZM-bl or MDMs to monitor the nuclear mobility of VRCs.

To measure the stability of VRC clusters in MDM nuclei, cells were co-infected with VSV-G pseudotyped pR9ΔEnv virus fluorescently labeled with either green (INmNG or INsfGFP) or red INmCherry for 24–72 h. The nuclei were labeled with Hoechst-33342 for 10 min and nuclear IN spots were visualized. PF74 (25 μM), CA-I (100 μM), TNFα (10 μg/ml), or 1,6-Hexanediol (10%) were added to cells on a microscope stage and co-trafficking of INmNG/INmCherry was monitored for varied time intervals ranging from 0.5 to 2 h by time-lapse live-cell imaging every 10 sec in the presence of drug. When indicated, cells were incubated for additional 24 h after initial time-lapse imaging and reanalyzed for the presence of VRC clusters.

**Image acquisition**. 3D time-lapse live-cell imaging was carried out on a Zeiss LSM880 AiryScan laser scanning confocal microscope, using a 63×/1.4NA oil-immersion objective. Live-cell imaging was performed using ×1 zoom, 512 × 512 frame size, 240 nm/pixel, and 1.5 μs pixel dwell time and bi-directional line scanning mode. To visualize uncoating and nuclear entry, 25 neighboring fields of view were imaged by tile scanning. Live-cell imaging was performed by acquiring 12-bit 512 × 512 image stacks (11–15 Z-stacks) spaced by 1 (40×/1.2 NA objective) or 0.7 μm (63×/1.4NA objective). Z-stacks were acquired every 5–7 min (slow acquisition) for 40 h (<480 time frames), when using a 405 nm laser line to excite

Hoechst-33342-stained nuclei in a 2-track line scanning mode. Alternatively, Z-stacks were acquired every 40–90 sec (fast acquisition) for 12–18 h (<1200 time frames), while using a less phototoxic SNAP-SiR647 labeled nuclear envelope marker in a single-track line scanning mode. Nuclei stained with Hoechst-333342 or SNAP-lamin and containing INmNG and CypA-DsRed were imaged using highly attenuated 405, 488, 561, and 633 nm laser lines corresponding to 0.2, 2.5, 1, and 1% laser power, respectively, and a pinhole was adjusted to 150 μm. The spectral detector bandwidth was adjusted to minimize spectral overlap between channels. Specific settings include: Hoechst-33342/AlexaFluor405 (415–470 nm), INmNG (490–558 nm), CypA-DsRed/AlexaFluor568 (572–625 nm), and SNAP-SiR647/AlexaFluor647/Cy5 (640–700 nm). The Z-Piezo stage and DefiniteFocus module (Carl Zeiss) were utilized to correct for axial drift. The above-mentioned imaging parameters were empirically determined to optimize imaging while minimizing fluorescence photobleaching and cellular toxicity.

When imaging fixed cells, 12-bit 1024 × 1024 pixel images were acquired using higher laser powers: 0.5% 405 nm, 5% 488 nm, 2% 561 nm, and 2% 633 nm, and 4× line averaging in order to improve signal-to-background ratio. We also used ×2 optical zoom, 0.07–0.14 nm/pixel, and 1.5 μs pixel dwell time and bi-directional line scanning mode. More stringent axial sampling (~45 Z-stacks spaced by 0.3 μm) was used.

For photoactivation experiments, TZM-bl SNAP-Lamin cells transiently expressing PA-C6 were used. A region of interest (ROI) was drawn in the center of the nucleus of uninfected cells or around nuclear INmCherry puncta in infected cells at 4 hpi. A 405 nm laser line was used at 10% power according to factory settings for ten iterations with a pixel dwell time of 1.5 μs to photo-activate PA-C6 in the ROI. Images pre- (five frames) and post-photoactivation were collected at a frequency 2.5 s/frame (single Z-stack) in uninfected cells to monitor PA-C6 mobility, or at 20 s/volume in the case of infected cells to monitor VRC co-mobility with PA-C6. To track the PF74-induced increase in nuclear IN puncta mobility (shown in Fig. 5e, f and Supplementary Movie 7), we used AiryScan-fast imaging modality (using 16-Airscan detectors) to image a single nuclei in a smaller 128 × 128 format. In this imaging modality, only the Hoechst-33342 (ex. 405 nm) and INmNG (ex. 488 nm) were imaged using the following parameters: 142 nm pixel size, zoom ×1.6, pixel dwell time 0.73 μs, Z-stacks spaced by 0.5 μm, and imaging frequency 2.5 s/volume for a period of 1 h. Airyscan images were processed using a proprietary 3D-AiryScan processing module in Zen software using Auto-thresholding. The 3D images were later converted to 2D maximum intensity projections for single-particle tracking using ICY software. 3D-image series were analyzed off-line using ICY image analysis software[65].

**Single-particle tracking and image analysis**. The initial annotation of HIV-1 uncoating and INmNG complex entry into the nucleus was done manually by examining time-lapse movies. After visual inspection, software-assisted single-particle tracking was used to analyze viral complexes in the cytoplasm, determine the time of arrival at the nuclear membrane, and the time of penetration/import into the nucleus. Single-particle tracking was performed using the ICY image analysis platform. Single Z-sections containing the object of interest were manually identified after examination of the whole volume, extracted, aligned in time, and used for single-particle tracking. 3D Z-stack images containing the single virus fluorescent puncta of interest were examined converted to 2D maximum intensity projections using Zen software and analyzed using ICY software. INmNG objects were detected using the wavelet Spot Detection plugin. After visual detection of INmNG spots, the objects were tracked using pre-defined parameters for a mixed diffusive and active transport motion model using the Spot-Tracking plugin. When adding drugs during live-cell imaging, sample displacements confounded continuous single-particle tracking. The broken single-particle trajectories were manually verified and stitched using the ICY Track Manager plugin. Cellular motion artifacts were corrected by tracking single nuclear volumes, using the Active-Cells plugin and subtracting the nucleus center trajectory from the single IN complex trajectory. The coordinates for the nucleus and single particles were visualized with Track Manager available in ICY[65] and exported into excel for further analysis. Single-particle intensity traces were normalized to initial fluorescence intensity at the time of initiation of image acquisition after local background subtraction (see below). The nuclear SNAP-Lamin signal was normalized by subtracting the background signal and setting the peak intensity at the mid-lamin section as 100%. Single docked cores at the nuclear membrane were identified manually based upon the apparent colocalization to the nuclear membrane and tracked using the ICY software, as described above.

In order to discriminate between cytoplasmic and nuclear IN spots, an in-house protocol was created using the ICY protocols module (refer to the ICY online tutorial). Briefly, the nuclear volume in three dimensions was detected using the lamin intensity by the HK-means and the connected components plugins in ICY[66]. To avoid detection of nuclear membrane-associated IN spots, the obtained three-dimensional ROI corresponding to the nuclear volume was shrunk by 0.5 μm in X–Y–Z using an ROI-erosion plugin. The IN complexes within the eroded ROI were considered as nuclear spots, while remaining spots were deemed to reside in the cytoplasm.

For object intensity analysis, the ROI of the detected fluorescent spots was dilated by one pixel in all dimensions to include the background pixels. The average intensity in the dilated one-pixel-thick background region was calculated

and subtracted from the average intensity of the original ROI followed by multiplying this resulting signal by the total voxels in the original ROI yielding background subtracted (BG) sum intensities of objects. The BG correction routine was used for all intensity-based analysis. The BG-corrected sum intensity of the INmNG, INmCherry, CA/p24, CypA-DsRed, CPSF6, EdU/vDNA, SC35, CDK9/pS175, CyclinT1 signals within each nucleus-associated VRCs was measured. The fraction of intranuclear IN spots that contained above-background levels of CA/p24, CypA-DsRed, EdU/vDNA, CPSF6, CyclinT1, CDK9/pS175 was considered colocalized with these markers. Where relevant, the intensities of virus and cellular proteins were analyzed after sorting SC35(+)/(−) IN spots (see below).

Speckle analysis was performed using an in-house ICY protocol[65]. Briefly, for all cell types (HEK293T/17 TZM-bl, Jurkat, CD4+ T cells, and MDMs) speckle volume in three dimensions was determined using the SC35 intensity within the nuclei detected using Lamin, Hoechst-33342, or CPSF6 staining by using the HK-means plugin[66]. A minimum size of 10 and a maximum size of 2000 pixels were chosen based on SC35-appearance in MDM and applied across all cell types. The speckle-picking routine was kept constant in order to avoid bias. The speckle detection results were confirmed by visual inspection. Note: this type of speckle analysis occasionally picked multiple closely located speckles in all cell types except MDMs. Because of this effect, the number of NSs in non-MDM cells is underestimated (Fig. 3c), while the individual speckle volumes are overestimated (Fig. 3d). For analyzing the association of IN puncta with SC35-positive compartments, the SC35 volumes (see below) were dilated by one pixel and used as a mask for IN puncta detection. All other nuclear IN spots were considered as SC35(−).

Note: all images shown in figures were subject to smooth filter (strength 4) available in Zen software (Zeiss). The same linear thresholding of gray values was applied across various conditions in each independent experiment. This image processing was used only for presentation purposes and was not applied prior to quantitative image analyses.

**Analysis of HIV-1 integration sites**. Chromosomal regions that lie within 500 nm of NSs are defined as SPADs[33]. The SPAD data set[33] was reproduced herein using Bowtie2[67] to map raw sequence reads from archived file SRR3538917 to human genome build hg19. As outlined in Chen et al.[33], SPAD sites were defined as TSA-Seq scores greater than the 95th percentile, which yielded 1,547,458 SPAD sequences each of 100 bp in length. SE sequences from Jurkat T cells were directly downloaded as a bed file from the dbSUPER database[68].

Illumina sequence reads from prior HIV-1 integration studies[19,22,35] were mapped to the human genome as previously described[19,69,70]. In brief, U5 vDNA sequences trimmed from Illumina read1 reads were deduplicated and aligned to hg19 by BLAT[71] or HISAT2[72]. Unique integration sites were selected for downstream analysis.

The RIC data set was determined by digesting hg19 with MseI and BglII restriction enzymes in silico. Percentages of bulk integration sites that fell within SPAD or SE sequences were calculated using bedtools intersect[73]. Associated $p$ values were calculated by Fisher's exact test in a pairwise manner using Python. The frequency of integration relative to SPADs in Fig. 8d–f and Supplementary Fig. 6b was plotted using 200 kb bins.

RIGs were identified as genes targeted for integration across cell types (HEK293T, HOS, MDM, and primary CD4+ T cells)[19]. Briefly, the number of integrations per RefSeq gene was calculated using bedtools[73] for wet bench libraries and the in silico generated RIC. In each cell type, integration frequency observed in individual genes was compared to that of the RIC to identify genes that are frequently targeted for integration (genic integration frequency > RIC and $p < 0.05$; Fisher's exact test). RIGs were then defined as genes frequently targeted for integration in at least three of the studied four cell types. This yielded a total of 46 RIGs from WT CPSF6-expressing cell types and 30 RIGs from cells infected under CPSF6-defective conditions (Supplementary Table S1). Distances from RIG to nearest SE or SPAD were determined as the distances between the closest boundaries between the gene vs. SE/SPAD using bedtools[73].

**Statistics and reproducibility**. Unless indicated otherwise, statistical significance was determined using a nonparametric Mann–Whitney rank-sum test. $p < 0.05$ (*) was considered significant; ** and *** denote $p < 0.01$ and $p < 0.001$, respectively. The number of experiments and error bars are indicated in the figure legends. Images in Fig. 5a, c, e, g are representative of 20, 45, 33, and >120 nuclei, respectively, from >3 independent experiments/donors (CD4+ T cells). Images in Fig. 8i are representatives of 45 nuclei from three donors. Images in Supplementary Figure 5e are representative of >120 nuclei from three independent experiments/donors (CD4+ T cells).

**Reporting summary**. Further information on research design is available in the Nature Research Reporting Summary linked to this article.

**Data availability**
Source data are provided with this paper in a Source Data file. The data that support the findings of this study are available on request from the corresponding authors A.C.F. and G.B.M. Source data are provided with this paper.

**Code availability**
No new custom-written codes were used to analyze the results, all analyses were performed using a free image analysis software package ICY. For SPAD analysis, raw sequences were aligned to hg19 using Bowtie2 v2.3.4.3 and Samtools v1.3.1 was used to create Bam files. The ranking of TSA-Seq reads to >95th percentile was done using RStudio v1.2.5001. The dbSUPER database was accessed via https://asntech.org/dbsuper/. For integration sites, raw sequences were aligned to hg19 using BLAT v35 and HISAT2 v2.1.0. The Bedtool intersect command was performed via Bedtools v2.27.1. Python v3.7.4 and its module scipy.stats were used for to perform Fisher's exact test. Source data are provided with this paper.

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

## Acknowledgements

We gratefully acknowledge the reagents received from the NIH AIDS Reagents Program. We are grateful to Dr. Mattia Lion (Harvard University), Dr. Karen Kirby (Emory University), Dr. Mamuka Kvaratskhelia (University of Colorado), and members of the Melikyan lab for critical reading of the paper and helpful comments. The authors also thank Dr. Vineet KewalRamani for discussing his unpublished results. We would like to thank Ms. Nadia Paylor, Hui Wu, and Dr. Satya Prakash Singh for technical assistance. This work was supported by NIH R01AI129862 grant to G.B.M., R01AI052014 (to A.N.E.), and AI148382, AI120860, and U54 AI150472 (to S.G.S.).

## Author contributions

A.C.F. and G.B.M. conceived the study and analyzed data. A.C.F., M.M., M.J.P., K.P.R., S.L., and P.R.T. performed experiments and analyzed data. P.K.S., V.A., and A.N.E. established SPAD and SE data sets and analyzed integration site sequencing data. A.C.F., G.B.M., A.N.E., and S.G.S. co-wrote the paper. All authors read and edited the paper.

## Competing interests

The authors declare no competing interests.
