## [Peer Review File · Nature Communications]

Reviewers' Comments:

Reviewer #1:

Remarks to the Author:

This study by Francis Answath and coworkers uses imaging-based approaches to make key observations regarding the nuclear trafficking of HIV-1 complexes. Specifically, they observe that following nuclear import, HIV-1 Viral Replication Complexes (VRCs) localize to nuclear speckles and that this association is dependent on the ability of the viral capsid to interact with the host factor CPSF6. This in turn drives integration into DNA domains present in these speckles (SPADs). These observations have the novelty and impact which make it appropriate to publish in a strong journal such as this. However, there are significant issues with the manuscript. First, although the methodology used throughout the manuscript is consistently technically impressive, there is very little consistency in the methods of analysis the authors use to form their conclusions throughout the manuscript and the reason for this is unclear. Second, the manuscript contains extensive data that isn't directly relevant to the key conclusions of the manuscript. Some of this extraneous data is technically impressive and will be of interest to the field, while other pieces of data are not extensive or compelling enough to support strong conclusions (although collectively they might represent the better part of another manuscript exploring events in MDMs.). In point of this fact, the two key figures in the paper are figures, in my opinion, are parts of figures 4-5, which demonstrate that VRCs localize to SPADS in a CPSF6 dependent fashion, and figure 7, which shows that this impacts integration in SPADs. These figures show, convincingly, that localization to and integration into SPADs occurs in many cell types, yet almost half of the manuscript focuses extensively on macrophage infection. The authors could remove these studies and I would still be generally supportive of acceptance, should the issues described below be resolved, as the primary observations regarding CPSF6 and SPAD localization are of considerable significance. However, there are inconsistencies in analysis and other issues which require resolution prior to publication.

With regard to analytical methodology, the authors use a combination of approaches to monitor viral VRCs, including CypA-DSred fusion, Integrase labels and Edu nucleotide incorporation. The authors also show, impressively in figure 3, that individual VRCs traffic to common locations in the nucleus, and therefore use the aggregate signal generated by these VRCs as an experimental readout. At key times in the paper, the authors change the way they make the point that VRCs localize to speckles in a confusing way. While a diversity of approaches is obviously a positive, they are changed in ways that aren't complementary and are generally confusing. For example, a key finding in the paper is the observation that the A77V CA does not localize to SC35+ speckles (Fig 4). This is made with Edu labelling, absent any corroboration with virus specific markers. Later in that figure, the authors plot the CPSF6/Integrase ratio to demonstrate that PF74 prevents speckle

localization, but there seem to be many more straightforward measurements that could demonstrate this point. In figure 3, which first rolls out this result, the text discussed “the majority” of particles localizing to speckles, but the data are plotted to show the intensity, not the number or percentage, of particles in these compartments. The most straightforward way to measure the result the authors seem to be observing is shown in figure 4H, which shows the percentage of VRCs in SP35+ compartments and show that this association is disrupted by PF74 (which competitively inhibits CPSF6 binding). This approach should be used throughout the paper. If merging of complexes in the nucleus make quantifying the number of complexes not demonstrative in some cases, aggregate intensity of IN-GFP would be appropriate, and this can be explained in the text. However, throughout the many experiments, the analytical methodology should not change without justification.

Other Concerns:

Fig 1: The authors state that nuclear movement and “comparatively fast import of the majority of complexes” confounded tracking, such that only 30% of complexes were analyzed. This is unclear and potentially concerning. It isn't clear to me if the authors suggesting that the timing after infection was variable such that some viruses were already in the nucleus when they began imaging the cells or that a subpopulation of viral particles exhibited import behaviors substantially different from those that they analyzed such that they entered the nucleus more rapidly during the acquisition period? This is a point that needs to be addressed. If there are indeed multiple nuclear import pathways, as suggested by other studies, this gating strategy may be unintentionally biased towards looking at one or a subset of these pathways. The authors should clarify this and acknowledge any caveats that may be associated with their approach.

Figure 3 is perhaps the most important figure of the paper. In Fig 3A, GFP and Cherry tagged VRCs end colocalize in the same sub-resolution nuclear complexes. This was a cool experiment and strongly supports the notion that there are common sites to which nuclear VRCs traffic. However, the manner in which this experiment is analyzed and discussed is confusing and seems likely to lose the average reader, if not expert reviewers. I'm not really sure what Fig 3B is telling me. The legend talks of colocalization, but only mCherry sum fluorescence seems to be plotted. Red and green dots are used here to represent mCherry signal in puncta that are SC35+ and SC35-? That could not be more confusing, given the use of GFP and cherry constructs in 3A. Perhaps the legend or Y axis label is off? Couldn't one plot the GFP and mCherry intensity on x and y respectively, the same way they do in Fig S2H? Based on what I can infer, that should show big differences between SC35+ and SC35- puncta and really drive the point home. The use of green and red in a figure to indicate SC35+/- should surely be avoided. The 3D rendering in figure 3C obscure one of the key observations in the paper. Making this point in clear, channel separated and merged images is crucial to the story. The 3D renderings

seem best suited to the supplemental data section.

The authors do something similar to this in Fig 4A/B, but instead of using the Integrase tag, they use Edu staining that does not seem to be colocalized with IN-GFP. Do the authors never observe any Edu signal in the absence of virus?

The observation that high doses of PF74 does not disrupt CA assemblies (and in fact may provide a mechanism by which to more accurately assess the amount of CA associated with a VRC in imaging studies) is in disagreement with a number of prior studies which suggest that high doses of PF74 disrupt CA assemblies. However, the data are quite strong (and we have also made similar observations). This should be acknowledged, albeit gently and dispassionately, in the discussion, as I think it will provide valuable clarity to the field and reassurance to other researchers who may be puzzled by similar observations, given the widespread belief that high doses PF74 causes potent capsid disassembly.

In the second section of the results, the authors make a distracting argument to defend on of their labelling methods Cyclophilin-DS-Red (CDR) and their previous conclusion that CA is lost from the VRC upon nuclear entry. In doing so, they inappropriately cite a number of studies. The authors state, "Since loss of CDR at the nuclear envelope correlates with loss of HIV-1 CA/p24 upon nuclear entry [13-15, 17, 19, 29-31], decrease in CDR signal is interpreted as HIV-1 uncoating." The vast majority of these references have nothing to do with this point and most used alternative approaches to measure uncoating. By citing them in this way, the authors seem to be trying to buttress against reviewer concerns associated with CDR being an indirect indicator of CA binding. The method isn't perfect, but I'm not aware of the method of HIV-1 labelling that is perfect. Moreover, just two paragraphs later, they state "Similar results were obtained for HIV-1 co-labeled with INmNG and CDR as a marker for CA (Fig. S2F, G), confirming that a subset of CA and CDR persists on viral complexes beyond nuclear entry."

The way the authors approach the observation that RT facilitates nuclear import is puzzling to me. The authors make the interesting observation that the half-life of nuclear import decreases by half when SAMHD1 is depleted by Vpx, the authors use this data to question previous studies that suggest that cytoplasmic reverse transcription impacts core stability and uncoating and support their prior work, concluding "vDNA synthesis is not required for HIV-1 nuclear import in HeLa cells or MDMs". They then state that this point is also supported by evidence that vDNA synthesis is "not affected by high-dose of PF74 (Fig 6E, F) which has been reported to block reverse transcription [31], perhaps by altering the stability of intact cores in the cytoplasm [13, 62], provides additional evidence that nuclear VRCs are not confined inside intact conical cores." However, 6E and F do not measure reverse transcription, and the logic that they cite as "additional evidence" is data that, as mentioned above, they demonstrate to be incorrect, as PF74 does NOT seem to

impact core stability (6C, 6F). If the authors were to take a more dispassionate view of the data, there are numerous explanations that might allow them to integrate their observations with the observations of others in a more unifying fashion, which I think they should consider. Minimally, they should remove the current language, which I think is a biased interpretation of their data that is focused on defending aspects of their prior studies.

Reviewer #2:

Remarks to the Author:

In their manuscript "HIV-1 replication complexes accumulate in nuclear speckles and integrate into speckle associated genomic domains", Ashwanth C. Francis et al. reports that viral replication complexes traffic to and accumulate within nuclear speckles and that these steps precede the completion of viral DNA synthesis. They describe that HIV-1 transport to nuclear speckles is dependent on the interaction of capsid protein with CPSF6, which is also required to stabilize the association of the viral replication complexes with nuclear speckles. Finally, they report that integration site analyses reveal a striking preference for HIV-1 to integrate into speckle-associated genomic domains.

This an extensive and well done analysis that largely confirms already established data that HIV integrates into actively transcribed genes. While the authors provide a new imaging-based method to understand viral trafficking/integration and their data correlates well with previous observations in the field, the novelty of their findings remains underdeveloped. In addition, more functional data connecting the imaging with the observed phenotypes would be beneficial.

Specific concerns

In figure 4 A-C, the authors present evidence that the capsid mutant A77V disrupts the Capsid-CPSF6 interaction. Follow-up studies are largely done with the PF74 compound instead of the A77V mutant, which appears unspecific and no viability data are provided. Experiments with PF74 should also be performed with the A77V mutant and viability data for PF74 should be provided.

In order to show that SC35 is required for viral localization, experiments knocking down SC35 should be performed.

It is unclear how the "Analysis of HIV-1 Integration Sites" was performed. Are these existing data sets? How were the data on CPSF6 +/- cells acquired? This part needs more detail.

Please explain in better detail the Edu stainings; why does it only label the viral

DNA?

Figure 6B. Please, further explore the exclusion of VRCs from the Nuclear Speckles and define if they relocate to the nucleoplasm or if they are degraded (e.g. use of proteasome inhibitors).

Minor concerns

Figure 2E. Please, include error bars or show individual donors.

Figure 3. If possible, also include other markers of nuclear speckles, as an alternative to SC35. Authors indicate that important α -Amanitin experiments have been performed but no data were found.

Figure 5C. Please also present the IN panels without merging with PA-C6 signal.

Figure 5D. Please present the average signal of different Nuclear Speckles, and not only a representative of one NS where the PA-C6 signal disappears. It appears in 5C that some of the PA-C6 still remains in speckles near the IN signal.

Figure 6G. Please, explain why you observe dotted vDNA speckles in all samples treated with 25 μ M PF74. Are the cells viable upon 25 μ M PF74 treatment?

Suppl. Fig. S3. Please, include representative images of the different treatment conditions of the Table. Specially for PF74 25 μ M.

Please carefully review writing. According to Figure 6G, H, prolonged exposure to 25 μ M PF74 did NOT affect vDNA synthesis in the nucleus.

Reviewer #3:

Remarks to the Author:

The authors present an imaging heavy study of HIV-1 behavior in the nucleus, especially in macrophages. The constructs and labeling systems are well established (viral replication complex that contains dual color fluorescent protein label, reverse transcription – RTC, pre-integration – PIC and RNP) and the question asked are hard if impossible to answer with a non-imaging approach. The overall design of the experiments seems sound, although I am not nearly as much a HIV (or TSA seq) expert as I am one in technical aspects of imaging.

Consistently I have only a few questions to the authors with regard to the HIV part of their work and assume other reviewers will be better positioned to give feedback.

- A relatively high MOI is used, which seems consequent with respect to the clustering questions asked, however, my understanding from discussions with HIV

experts is that a MOI of 1 is the most (if not only) relevant form of infection in human. Is there support for MOI's this high in macrophages? Could a bit more background be presented to position this choice? Or is the assumed scenario for this study such that the MOI is justified otherwise?

- The conclusion at the end of page 6 of having shown a "default pathway" seems a bit to big – it seems to me that more biochemical studies would be needed to establish such a fundamental statement across multiple cell types.

- For the PF74 experiment, what is brief? If I wanted to reproduce this experiment, I would have a hard time to guess what you did exactly and with what kind of variance.

- I appreciate the pull down experiments, they strongly support your complex formation conclusions.

I do have some problems following the imaging protocols and analysis steps taken, and am missing significant information on image processing for the provided figures and supplemental material. I say so after spending quite some time trying to align the described z-stack time series with the presented time points in the movies.

- It would help if it could be indicated clearly which experiment was run on which microscope platform.

- From the provided information it is unclear to me what effective sampling size [nm per pixel] was used and if magnification was fixed to the optical mag of the objective or if additional settings were optimized.

- Laser power in % is not interpretable to me and is also only provided for the 405 nm line. What is the variance day to day? Over what time frame were experiments conducted, were detectors calibrated at all?

- I am still trying to make a best guess if a typical z-stack contained 45 planes or 45 z-stacks were taken, if 40-90 sec was the imaging duration of imaging frequency for stacks (respectively 5-7 minutes) why these speeds were sufficient to track particles (in multiple movies it is very hard to see any signal under the lines that were superimposed) and how 2.5 frames per second relate to 20s/ stack (different number of planes?) and if the 20s are the imaging duration or time between imaging events.

- Was the pixel dwell time generally 1.5 microseconds or just for one imaging condition?

- Given the long time periods, if I understand correctly over 40 hours and more) it would be informative to know if lateral stage drift was assessed, or even corrected for and how.

- Given the different N.A.s and refractive index mismatches, were R.I. of media controlled, what was the same resolution achieved on both systems? Was collection efficiency tested?

- I am missing information on image registration. How were axial offsets measured and corrected, how large were they? Same question for lateral aberrations.

- What was the width chosen for the emission filter (if applicable) or the spectral detector settings?

- What gain was used and how was the setting established? Was linearity checked?
- In several movies the jump distance between presented time points was bigger than the diffraction limited, sometime substantially bigger, what is the proof (or rational) to assume the particle identity is known?

My confusion for the image processing part is even bigger:

- What was done during the 3D image processing? It is unclear if linear adjustments for presentation were made (okay to do if documented) and if the processing included any filtering (as the data presented might suggest, unless the code and compression caused a bluer), was any registration between channels applied?
 - What are "2D single particle tracking parameters"? What distance and trace filling values were chosen, and why/based on which control?
 - How is 5% "initial intensity" defined? Was background subtracted, was signal based on an area threshold or as integral from a multiparameter fit? If so, was the fit width consistent with the imaging parameters? What fit model was used? Was bleach correction applied, if so, based on which model, if not how was the signal interpretation adjusted to distinguish bleaching from particle decay?
 - What was the criteria for a signal to be interpreted as single core? For CA/p24 some imaging parameters are given in the analysis section, but the time information is missing.
 - Lamin signals are frequently not continuous and somewhat blurred. How was that handled in the analysis? Was dilation used generally and always with the same parameters? How many rounds were applied?
 - Were SC35 compartments really varying in size between 10 and 2000 pixel? And were pixels of the same size?
 - How were thresholds established? Did the data show plateaus (settings were changing the thresholding parameter slightly had no or little impact on the results)?
- As a summary, I am not able to reproduce any of your analysis, you are not giving enough information.

The supplemental movies are very hard to interpret, OME-Tiff (or even Tiff) images would help as it seems you combined still images and used the copy number of the images to impact the play time of the movie. Extracting the stills from the movies I have trouble seeing the signal in some cases, but it is possible that a faint signal would be lost in the diverse format conversions. In general a collage of images that document a trace (with the z-value being indicated if 3D tracking is done) would be very helpful.

On the statistical analysis, was the Mann-Whitney test the only applicable? Why would a two-sided ANOVA not fit your data?

I hope my questions allow you to improve your methods section and that other reviewers were able to comment more on the individual experiments. Being an imaging person myself I am thrilled to see your work!

David Grunwald, UMass Medical School, RNA Therapeutics Institute

Reviewer #1:

...there is very little consistency in the methods of analysis the authors use to form their conclusions throughout the manuscript and the reason for this is unclear.

A: We appreciate the comment and have made substantial changes in the manuscript. Please see specific responses below.

Second, the manuscript contains extensive data that isn't directly relevant to the key conclusions of the manuscript. Some of this extraneous data is technically impressive and will be of interest to the field, while other pieces of data are not extensive or compelling enough to support strong conclusions (although collectively they might represent the better part of another manuscript exploring events in MDMs.). In point of this fact, the two key figures in the paper are figures, in my opinion, are parts of figures 4-5, which demonstrate that VRCs localize to SPADS in a CPSF6 dependent fashion, and figure 7, which shows that this impacts integration in SPADs. These figures show, convincingly, that localization to and integration into SPADs occurs in many cell types, yet almost half of the manuscript focuses extensively on macrophage infection. The authors could remove these studies and I would still be generally supportive of acceptance, should the issues described below be resolved, as the primary observations regarding CPSF6 and SPAD localization are of considerable significance. However, there are inconsistencies in analysis and other issues which require resolution prior to publication.

A: We very much appreciate the critique and can see that focus on MDMs in the context of HIV-1 trafficking to and integration within nuclear speckles (NSs) might seem excessive. While we feel that the MDM-related results are central to this study, we removed less essential data pertaining to the kinetics of vDNA synthesis, nuclear import and integration (former Fig. 2, Fig. 4A-C, Suppl. Fig. 1E, F, Suppl. Fig. 2I-K, Suppl. Fig. 4K, and Suppl. Fig. 6). We also have extensively revised the text to improve clarity. We agree that the main findings of our study are indeed the VRC transport to nuclear speckles and integration in SPADs. However, this study also addresses discrepant findings on HIV-1 uncoating in MDMs reported previously. In spite of the slow rate of HIV-1 reverse transcription in MDMs, results of single particle imaging suggested that, similar to cell lines, loss of the conical core integrity occurs within minutes after viral fusion¹. In contrast, other imaging-based studies concluded that, whereas CA is largely lost upon HIV-1 nuclear import in HeLa cells²⁻⁵, little or no CA loss occurs upon nuclear import in MDMs^{5,6}. Our study addresses this question by: (1) presenting evidence that a large portion of HIV-1 CA is lost prior to nuclear import and (2) demonstrating that the reason others have observed strong CA signals in the MDM nucleus was the merger of multiple VRCs. Another reason for our initial focus on MDMs is that, by virtue of containing only a few very large NSs, these cells are ideally suited for visualization of VRC merger (Figs. 1-3). Perhaps owing to a large size of NSs in MDMs, multiple VRCs can form extremely stable clusters that could not be readily displaced from NSs by PF74 (Fig. 6) or dispersed by harsh treatment involving detergent and sonication (Suppl. Fig. 3). Experiments in MDMs form the basis of our investigations. We used other cell types to confirm the existence of a default pathway of HIV-1 transport and accumulation in nuclear speckles.

At key times in the paper, the authors change the way they make the point that VRCS localize to speckles in a confusing way. While a diversity of approaches is obviously a positive, they are changed in ways that aren't complementary and are generally confusing. For example, a key finding in the paper is the observation that the A77V CA does not localize to SC35+ speckles (Fig. 4). This is made with Edu labelling, absent any corroboration with virus specific markers.

A: In terminally differentiated non-cycling MDMs, the EdU nucleotide is not incorporated into the nuclear DNA, but incorporated into the vDNA upon reverse transcription. In the absence of vDNA synthesis (nevirapine treatment), very rare EdU spots are observed in the nucleus, as has been previously demonstrated by other groups⁵⁻⁷. Our results strongly support the lack of cellular DNA staining by EdU, as shown in Fig. 1e, Fig. 7c, d and Fig. 8i, k, as well as in the Suppl. Fig. 2c-h. Given a strong colocalization of INmNG and EdU spots in the MDM nucleus (e.g., Fig. 1e), we opted to use the latter signal to visualize VRCs in the former Fig. 4A. However, following the reviewer's suggestion, we replaced this figure with a new Figure 2c-f that shows IN-labeled WT and A77V VRC colocalization with NSs and the nuclear membrane.

Later in that figure, the authors plot the CPSF6/Integrase ratio to demonstrate that PF74 prevents speckle localization, but there seem to be many more straightforward measurements that could demonstrate this point. In figure 3, which first rolls out this result, the text discussed "the majority" of particles localizing to speckles, but the data are plotted to show the intensity, not the number or percentage, of particles in these compartments. The most straightforward way to measure the result the authors seem to be observing is shown in figure 4H, which shows the percentage of VRCs in SP35+ compartments and show that this association is disrupted by PF74 (which competitively inhibits CPSF6 binding). This approach should be used throughout the paper. If merging of complexes in the nucleus make quantifying the number of complexes not demonstrative in some cases, aggregate intensity of IN-GFP would be appropriate, and this can be explained in the text. However, throughout the many experiments, the analytical methodology should not change without justification.

A: We agree that insufficient justification was provided for analyses shown in Fig. 4. The ratio of CPSF6/IN signals was plotted to show CPSF6 accumulation at speckle-associated VRCs, not just to demonstrate the effect of PF74. Raw INmNG and CPSF6 fluorescence is plotted in Suppl. Fig. 5a-d. As the reviewer pointed out, the tendency of VRCs to merge in the MDM nucleus renders severely limits the utility of quantifying the number of complexes and their colocalization. We therefore rely on the INmNG or INmCherry intensity information to assess accumulation of VRCs in NSs. Per the reviewer's suggestion, we now show a correlation between the INmNG and INmCherry signals in the SC35(+) compartments compared to a lack of correlation elsewhere in the nucleoplasm (Fig. 2b), along with % VRCs in speckles for all cell types examined (Fig. 3b).

Other Concerns:

1. Fig 1: The authors state that nuclear movement and "comparatively fast import of the majority of complexes" confounded tracking, such that only 30% of complexes were analyzed. This is unclear and potentially concerning. It isn't clear to me if the authors suggesting that the timing after infection was variable such that some viruses were already in the nucleus when they began imaging the cells or that a subpopulation of viral particles exhibited import behaviors substantially different from those that they analyzed such that they entered the nucleus more rapidly during the acquisition period? This is a point that needs to be addressed.

A: No nuclear VRCs were observed when imaging commenced at 30 min post-infection. Nuclear import of VRCs was observed at later times post-infection, but only 30% of nuclear IN puncta could be traced back to the nuclear membrane. All these tracked particles lost CypA-DsRed prior to nuclear translocation. The remaining 70% of nuclear INmNG puncta could not be reliably back-tracked. This is because, in addition to MDM migration, their nuclei often underwent lateral, axial and rotational movements, which limited our ability to quantify the interaction of VRCs with the nuclear membrane, their residence times and subsequent nuclear

import. This is in contrast to less mobile TZM-bl cells in which we tracked nearly all IN puncta entering the nucleus ². It thus appears that failure to track the majority of particles in MDMs is related to more pronounced movement of their nuclei, as well as to single particle crowding at the nuclear membrane of these cells. We now acknowledge these limitations in the revised manuscript:

“A substantial lateral and axial movement of the MDM nuclei greatly limited our ability to reliably track the nuclear entry step of the majority of VRCs. As a result, we were able to track nuclear envelope docking and uncoating steps of a subset (~30%) of VRCs that entered the nucleus of less mobile cells. For these particles, we consistently observed loss of CDR signal at the nuclear membrane, prior to nuclear import (Fig. 1a, b, Suppl. Fig. 2a, b and Suppl. Videos 1 and 2), similar to uncoating observed in HeLa-derived TZM-bl cells”.

2. If there are indeed multiple nuclear import pathways, as suggested by other studies, this gating strategy may be unintentionally biased towards looking at one or a subset of these pathways. The authors should clarify this and acknowledge any caveats that may be associated with their approach.

A: We would like to refrain from commenting on the existence of multiple nuclear entry pathways for HIV-1, since our assay does not really address this point.

3. Figure 3 is perhaps the most important figure of the paper. In Fig 3A, GFP and Cherry tagged VRCs end colocalize in the same sub-resolution nuclear complexes. ... However, the manner in which this experiment is analyzed and discussed is confusing and seems likely to lose the average reader, if not expert reviewers. I'm not really sure what Fig 3B is telling me. The legend talks of colocalization, but only mCherry sum fluorescence seems to be plotted. Red and green dots are used here to represent mCherry signal in puncta that are SC35+ and SC35-? That could not be more confusing, given the use of GFP and cherry constructs in 3A. Perhaps the legend or Y axis label is off? Couldn't one plot the GFP and mCherry intensity on x and y respectively, the same way they do in Fig S2H? Based on what I can infer, that should show big differences between SC35+ and SC35- puncta and really drive the point home. The use of green and red in a figure to indicate SC35+/- should surely be avoided. The 3D rendering in figure 3C obscures one of the key observations in the paper. Making this point in clear, channel separated and merged images is crucial to the story. The 3D renderings seem best suited to the supplemental data section.

A: The reviewer's point is well-taken, especially in regards to the use of colors in the former Fig. 3B. We re-plotted this figure (currently Fig. 2b) to show a correlation between the intensity of INmNG and INmCherry puncta in SC35-positive compartments and avoided green and red colors. The 3D-rendered images in the current Fig. 3A allow one to readily appreciate the volumes of the nuclear speckles in different cell types, and especially the complex shape of NSs in MDMs. The HIV-occupied nuclear speckles are highlighted in grey. We believe that this visualization helps the reader to quickly assess the VRC colocalization and differences in speckle size and numbers in different cell types, which are analyzed in current Fig. 3c-e (former Fig. 3E-G). Channel-separated and merged images are shown elsewhere in the manuscript (see, for example, Fig. 4a, b for speckle association).

4. The authors do something similar to this in Fig 4A/B, but instead of using the Integrase tag, they use Edu staining that does not seem to be colocalized with IN-GFP. Do the authors never observe any Edu signal in the absence of virus?

A: Please see our response to this point above. Fig. 1e, Suppl. Fig. 2c-f, Fig. 7c, d and Fig. 8i-k show excellent colocalization of Edu and INmNG signals in the nucleus, except for a small fraction of small IN puncta that did not exhibit detectable Edu signal. In these figures, nearly no Edu puncta are seen in the nucleus. On a few occasions, we observed diffuse Edu/vDNA signal not colocalized with IN (Fig. 7c) at 72 hpi or later. Thus, Edu staining reliably identifies nuclear VRCs in MDMs. The choice of Edu/vDNA labeling in the former Fig. 4A, B was motivated by the desire to test if a virus not labeled with INmNG will also colocalize with nuclear speckles (SC35+) in infected eGFP-expressing MDMs. As pointed out above, in the presence of Nevirapine, we and others rarely (<0.2 Edu puncta/nucleus) observed distinct Edu puncta in MDM nuclei that is not associated with IN-labeled VRCs. In the revised manuscript, we removed the Edu results and replaced these with Figure 2c-f showing IN-labeled WT and A77V VRC colocalization with NSs and the nuclear membrane, respectively.

5. The observation that high doses of PF74 does not disrupt CA assemblies (and in fact may provide a mechanism by which to more accurately assess the amount of CA associated with a VRC in imaging studies) is in disagreement with a number of prior studies which suggest that high doses of PF74 disrupt CA assemblies. However, the data are quite strong (and we have also made similar observations). This should be acknowledged, albeit gently and dispassionately, in the discussion, as I think it will provide valuable clarity to the field and reassurance to other researchers who may be puzzled by similar observations, given the widespread belief that high doses PF74 causes potent capsid disassembly.

A: We agree that the effects of PF74 on the HIV-1 core stability are complex, perhaps accounting for discordant published results. In this regard, the recent work from the Boecking lab ⁸ is interesting. Through sensitive single particle measurements of loss of core integrity (release of iGFP) and uncoating (using a CypA), they have concluded that high doses of PF74 compromise the core integrity, but stabilize the remaining hexameric CA lattice. We have previously reported the stabilizing effect of 2-10 μ M PF74 on HIV-1 core ^{9, 10}. Here, we show that a high-dose of PF74 (25 μ M) stabilizes cores located in the cytoplasm, but not VRCs in the nucleus. We expanded the discussion of PF74 effects in the manuscript:

“Interestingly, in contrast to a recent report ¹¹, we did not observe reduction in CA signal associated with nuclear VRCs upon PF74 treatment of infected HeLa cells (Fig. 6c and Supplementary Fig. 5g, h). In contrast, we observed a selective enhancement of nuclear VRC-associated CA signal by PF74 compared to cytoplasmic complexes (Supplementary Fig. 5g, h), highlighting a fundamental difference in the structure and/or composition of these complexes. We also note that the nuclear CA signal was 2-fold lower than that of the cytoplasmic cores (Supplementary Fig. 5g, h), suggesting that a fraction of CA molecules is lost upon nuclear import of VRCs. We therefore surmise that the reported nuclear CA-destabilizing effect of PF74 ¹¹ may be related to the use eGFP-CA fusion protein for HIV-1 core labeling. Moreover, in MDMs, where multiple VRCs form stable clusters (Supplementary Fig. 3), a high-dose of PF74 added after cluster formation had no effect on VRC cluster-associated vDNA synthesis (Fig. 7c, d). Because a high-dose of PF74 has been reported to block reverse transcription when added early during infection of HeLa cells ¹², by virtue of altering virus core-stability, this somewhat surprising result suggests that PF74 may not affect reverse transcription outside the context of an intact conical core.”

6. In the second section of the results, the authors make a distracting argument to defend on of their labelling methods Cyclophilin-DS-Red (CDR) and their previous conclusion that CA is lost from the VRC upon nuclear entry. In doing so, they inappropriately cite a number of studies. The

authors state, “Since loss of CDR at the nuclear envelope correlates with loss of HIV-1 CA/p24 upon nuclear entry [13-15, 17, 19, 29-31], decrease in CDR signal is interpreted as HIV-1 uncoating.” The vast majority of these references have nothing to do with this point and most used alternative approaches to measure uncoating. By citing them in this way, the authors seem to be trying to buttress against reviewer concerns associated with CDR being an indirect indicator of CA binding. The method isn't perfect, but I'm not aware of the method of HIV-1 labelling that is perfect. Moreover, just two paragraphs later, they state “Similar results were obtained for HIV-1 co-labeled with INmNG and CDR as a marker for CA (Fig. S2F, G), confirming that a subset of CA and CDR persists on viral complexes beyond nuclear entry.”.

A: The reviewer gives us too much credit, as we did not have an agenda when citing these papers. Our intent was to simply acknowledge the contribution of other groups for their detection of nuclear CA (not CDR). Nonetheless, we have removed these citations and the entire sentence from the revised manuscript. We consistently observe a major loss of CDR at the nuclear envelope, but also detect residual CA and CDR signals associated with nuclear VRCs, especially when using fixed cell imaging that yields a greater signal to background ratio. Suppl. Fig. 2c, d, g, h and i shows that both p24 and CDR are associated with nuclear VRCs, thus arguing against the possibility that loss of CDR at the nuclear membrane observed in our live cell experiments is a result of its displacement from the viral core.

7. The way the authors approach the observation that RT facilitates nuclear import is puzzling to me. The authors make the interesting observation that the half-life of nuclear import decreases by half when SAMHD1 is depleted by Vpx, the authors use this data to question previous studies that suggest that cytoplasmic reverse transcription impacts core stability and uncoating and support their prior work, concluding “vDNA synthesis is not required for HIV-1 nuclear import in HeLa cells or MDMs”.

A: We are sorry for misunderstanding. We do not question the notion that reverse transcription facilitates uncoating. In fact, we have observed this previously⁹. We also do not question the fact that HIV core loses integrity early in infection. The observation that HIV-1 complexes retain the ability to enter the nucleus of HeLa-derived cells in the absence of vDNA synthesis has been made by others^{13, 14} and we have confirmed this finding in cell lines². The lack of effect of reverse transcription inhibitors on nuclear import indicates that, besides vDNA synthesis, other events, such as contact with the nuclear pore components, may play a role in HIV-1 uncoating. Here, we extend the previous observation that vDNA synthesis is not required for nuclear import in MDMs. No claims were made regarding the effect of reverse transcription on the core stability in these cells. In fact, our unpublished results indicate that Vpx(+) treatment accelerates early uncoating of HIV-1 in MDMs. As was stated in Discussion of the original manuscript, the reasons for the accelerated nuclear import in SAMHD1 depleted MDMs is currently unclear. We know that Vpx degrades SAMHD1, thereby increasing the dNTP pool in MDMs and facilitating HIV-1 reverse transcription. As per the reviewer's suggestion, we removed these data from the current manuscript and will write a follow up paper on the subject.

8. They then state that this point is also supported by evidence that vDNA synthesis is “not affected by high-dose of PF74 (Fig 6E, F) which has been reported to block reverse transcription [31], perhaps by altering the stability of intact cores in the cytoplasm [13, 62], provides additional evidence that nuclear VRCs are not confined inside intact conical cores.” However, 6E and F do not measure reverse transcription, and the logic that they cite as “additional evidence” is data that, as mentioned above, they demonstrate to be incorrect, as PF74 does NOT seem to impact core stability (6C, 6F). If the authors were to take a more dispassionate view of the data, there are numerous explanations that might allow them to

integrate their observations with the observations of others in a more unifying fashion, which I think they should consider. Minimally, they should remove the current language, which I think is a biased interpretation of their data that is focused on defending aspects of their prior studies.

A: We apologize for incorrectly referencing the figure. The correct reference for vDNA synthesis in the nucleus is Fig. 7c, d (formerly Fig. 6G-I). This figure shows vDNA synthesis in the nucleus after addition of PF74 that blocks virus nuclear import. EdU is added with or without drugs (Nevirapine or PF74) at 24 hpi and, upon further incubation for 3 days, EdU incorporation into nuclear VRCs is determined. The data clearly show that vDNA synthesis can continue in the nucleus of MDMs, even in the presence of a high dose of PF74 added at 24 hpi. Since even lower concentrations of PF74 block virus nuclear import in MDMs when added at 24 hpi, increased vDNA signal in nuclear VRCs originates from reverse transcription occurring in the nucleus. This notion is further supported by the lack of vDNA synthesis (beyond what was already completed by 24h) in Nevirapine-treated samples. Collectively, these results suggest that, in MDMs, the addition of high-doses of PF74 after 24 hpi does not block reverse transcription (Fig. 7c, d) or infectivity (Fig. 7e) of nuclear VRCs. This is in contrast to the PF74 effect on HIV-1 cores in the cytoplasm when the drug was added at the onset of infection in HeLa-derived cells ⁴. We surmise that addition of a higher concentration of PF74 at time 0 inhibits reverse transcription and also blocks nuclear import, perhaps through independent mechanisms that involve intact cores, whereas the effects of PF74 on nuclear VRCs are clearly distinct. Once in the nucleus of MDMs, HIV-1 complexes are relatively resistant to a high-dose of PF74, as they continue to incorporate EdU. As pointed out above, we discuss these points in the revised manuscript, as stated above (see the response to point 5).

Reviewer #2:

While the authors provide a new imaging-based method to understand viral trafficking/integration and their data correlates well with previous observations in the field, the novelty of their findings remains underdeveloped. In addition, more functional data connecting the imaging with the observed phenotypes would be beneficial.

A: We respectfully disagree with the reviewer's assessment of the novelty of our results. The following new observations reported here have not been previously published.

- a. Live-cell visualization of single HIV-1 uncoating and nuclear import in MDMs.
- b. Demonstration of merger of HIV-1 complexes in the nucleus of MDMs (Fig. 1; Fig. 2; Suppl. Fig. 2).
- c. Discovery of a previously unknown default targeting of HIV-1 nuclear complexes to nuclear speckles in multiple cell types (Figs. 2 and 3).
- d. Demonstration of highly stable clusters formed by multiple VRCs in MDM NSs, which could not be separated after detergent extraction and sonication (Suppl. Fig. 3)
- e. Demonstration that the CPSF6-CA interaction is essential for HIV-1 transport to nuclear speckles and for VRC retention in these compartments.
- f. Visualization of selective recruitment of CPSF6 to VRCs residing in nuclear speckles.
- g. Discovery of an architectural basis for the HIV-1 integration site preference through integration in SPADs, which was not demonstrated before.
- h. Finding that the previous interpretation of a robust CA signal of HIV-1 complexes in MDMs as the lack of uncoating^{6, 15} was incorrect. We show that, in MDMs, merger of multiple VRCs increases the overall CA signal above the levels detected in single intact viruses.

As to functional validation, we related the observed nuclear clusters, the effects of drug treatment or the appearance of nuclear vDNA puncta to infection throughout the paper. The following functional results are presented: (1) vDNA synthesis in NSs-localized VRCs in MDMs (Fig. 7c, d); (2) loss of infection after displacing VRCs from NSs in TZM-bl cells by PF74 treatment (Fig. 6a, b, e); (3) relevance of HIV-1 localization in NSs to subsequent integration into SPADs (Fig. 8a-h); and (4) colocalization of nuclear VRCs with HIV-1 transcription sites (Fig. 8i-l). Collectively, our results show that NSs provide the architectural basis for compartmentalization of virus integration.

Specific concerns

1. In figure 4 A-C, the authors present evidence that the capsid mutant A77V disrupts the Capsid-CPSF6 interaction. Follow-up studies are largely done with the PF74 compound instead of the A77V mutant, which appears unspecific and no viability data are provided. Experiments with PF74 should also be performed with the A77V mutant and viability data for PF74 should be provided.

A: We stated: "The A77V mutation in CA, which compromises the CA-CPSF6 interaction but only marginally impedes infection of MDMs^{6, 16}, results in peripheral localization of nuclear HIV-1 complexes^{6, 17}." Here, A77V was used as a control to make the case that VRC transport to nuclear speckles is blocked in the absence of CA/CPSF6 interactions. After confirming this in the former Fig. 3A-C, we moved forward to investigate the nature of CPSF6 interaction with speckle-associated VRCs, which can only be done in the context of WT virus, using drugs like PF74. Additional experiments with A77V will not provide significant information because: (1) this mutant CA does not associate with CPSF6 at the nuclear pore^{6, 15}, (2) the mutant does not

traffic to nuclear speckles (current Fig. 2c-f), and (3) A77V does not exhibit considerably attenuated infectivity. We now include MDM viability data after prolonged treatment with PF74 (new Suppl. Fig. 7).

2. In order to show that SC35 is required for viral localization, experiments knocking down SC35 should be performed.

A: We apologize for the confusion. SC35 is just a marker for nuclear speckles. It is not required for HIV-1 localization to speckles. We therefore did not pursue knock down of this protein.

3. It is unclear how the “Analysis of HIV-1 Integration Sites” was performed. Are these existing data sets? How were the data on CPSF6 +/- cells acquired? This part needs more detail.

A: As stated in the Methods, we analyzed several previously reported integration datasets for association with speckle-associated genomic domains (SPADs). These previously reported integration datasets were critically important to the work because they allowed us to analyze the *de facto* propensity for HIV-1 to integrate into SPAD regions in a variety of cell types, including model HEK293T cells¹⁸, as well as primary MDMs¹⁹ and CD4+ T cells¹⁷. Moreover, these prior datasets allowed us to specifically parse the roles of known virus-binding integration cofactors LEDGF/p75 and CPSF6 in SPAD-targeting, as they harbored integration sites from factor specific knockout cells¹⁸ or from primary cells infected with CPSF6 binding defective CA mutant viruses^{17, 19}. The SPAD dataset was rebuilt in-house, as described in the Methods, using sequences deposited by the Belmont labs²⁰. The methods section has been expanded as follows:

“Chromosomal regions that lie within 500 nm of NSs are defined as SPADs²⁰. The SPAD dataset²⁰ was reproduced herein using Bowtie2²¹ to map raw sequence reads from archived file SRR3538917 to human genome build hg19. As outlined in Chen et al. 2018²⁰, SPAD sites were defined as TSA-Seq scores greater than the 95th percentile, which yielded 1,547,458 SPAD sequences each of 100 bp in length. SE sequences from Jurkat T cells were directly downloaded as a bed file from the dbSUPER database²².

Illumina sequence reads from prior HIV-1 integration studies^{17, 18, 23} were mapped to the human genome as previously described^{17 24, 25}. In brief, U5 viral DNA sequences trimmed from Illumina read1 reads were deduplicated and aligned to hg19 by BLAT²⁶ or HISAT2²⁷. Unique integration sites were selected for downstream analysis.

The RIC dataset was determined by digesting hg19 with MseI and BglII restriction enzymes *in silico*. Percentages of bulk integration sites that fell within SPAD or SE sequences were calculated using bedtools intersect²⁸. Associated P values were calculated by Fisher’s exact test in a pairwise manner using Python. The frequency of integration relative to SPADs in Fig. 8d-f and Supplementary Fig. 6b were plotted using 200 kb bins.

RIGs were identified as genes targeted for integration across cell types (HEK293T, HOS, MDM, and primary CD4+ T cells)¹⁷. Briefly, the number of integrations per RefSeq gene was calculated using bedtools²⁸ for wet bench libraries and the *in silico* generated RIC. In each cell type, integration frequency observed in individual genes was compared with that of the RIC to identify genes that are frequently targeted for integration (genic integration frequency > RIC and P <0.05; Fisher’s exact test). RIGs were then defined as genes frequently targeted for integration in at least 3 of the studied 4 cell types. This yielded a total of 46 RIGs from WT CPSF6-expressing cell types and 30 RIGs from cells infected under CPSF6-defective conditions

(Supplementary Table S1). Distances from RIG to nearest SE or SPAD was determined using bedtools²⁸

4. Please explain in better detail the Edu stainings; why does it only label the viral DNA?

A: MDMs are terminally differentiated cells that do not synthesize DNA. Therefore these cells are best suited for analyzing reverse transcription-mediated EdU incorporation into vDNA⁵⁻⁷. The virtual lack EdU/vDNA puncta in infected cells treated with nevirapine further supports the validity this approach to detect vDNA (e.g., current Fig. 7c, d). We did acknowledge that occasionally a diffused background EdU staining may be observed. This diffused staining is not virus specific. We now provide a more detail description as to why EdU selectively stained vDNA in viral complexes:

“vDNA was visualized by infecting MDMs in the presence of the nucleoside analog 5'-ethynyl deoxyuridine (EdU). Robust colocalization of EdU and INmNG signals was observed at 24 hpi (Suppl. Fig. 2c, d). The virtual absence of EdU signal in the MDM nuclei in the absence of reverse transcription (Nevirapine treatment, Suppl. Fig. 2e, f) demonstrates the lack of considerable DNA synthesis in these terminally differentiated cells.”

5. Figure 6B. Please, further explore the exclusion of VRCs from the Nuclear Speckles and define if they relocate to the nucleoplasm or if they are degraded (e.g. use of proteasome inhibitors).

A: This is an excellent suggestion. In fact, we have tried to block proteasomal activity with MG132 and Lactacystin. However, these drugs failed to prevent loss of VRCs that were re-located to the nucleoplasm after a brief treatment with 25 uM PF74. We therefore feel that further exploration of the mechanism of VRC loss, although interesting, is outside the scope of this paper.

Minor concerns:

6. Figure 2E. Please, include error bars or show individual donors.

A: As per the Reviewer 1's request this figure has been removed from the current manuscript.

7. Figure 3. If possible, also include other markers of nuclear speckles, as an alternative to SC35. Authors indicate that important α -Amanitin experiments have been performed but no data were found.

A: It is worth pointing out that SC35 is the gold standard for labeling nuclear speckle compartment. Several other proteins that localize to nuclear speckles are also confirmed by SC35 staining^{20, 29}. We now included a new Suppl. Fig. 4a, b that shows excellent colocalization of IN spots with NS stained for the alternative marker SON protein. The data for α Amanitin experiments is reported as a summary in Suppl. Fig. 3 in a form of a table. Treatment with this drug did not break pre-formed INmNG+INmCherry clusters in the nucleus of MDMs, even after prolonged treatment. The table simply reports this negative result. We included this information to support the observation that nuclear HIV-1 clusters found in MDMs (where HIV-1 replicates poorly in the presence of SAMHD1³⁰, former Suppl. Fig. 1B-E) are highly stable structures that could not be disassembled even after harsh treatments with drugs (Suppl. Fig. 3a) or detergents and sonication (Suppl. Fig. 3b-d).

8. Figure 5C. Please also present the IN panels without merging with PA-C6 signal.

A: While this would be helpful, we opted not to show separate IN panels due to space limitations on that busy figure. We show an example of this here for the reviewer.

9. Figure 5D. Please present the average signal of different Nuclear Speckles, and not only a representative of one NS where the PA-C6 signal disappears. It appears in 5C that some of the PA-C6 still remains in speckles near the IN signal.

A: As stated in the figure legend, there is no nuclear speckle staining in Fig. 5C, D (current Fig. 5). What we show is a colocalization of PA-CPSF6 with IN-labeled VRCs in the nucleus of living cells, their co-trafficking and loss of PA-CPSF6 association with VRCs after PF74 addition. As can be seen in Fig. 5c, all nuclear IN puncta (green) lose CPSF6 (red) signal. The track of a single IN complex losing PA-CPSF6 after PF74 treatment is thus shown. Aggregates of PA-CPSF6 (red) in the nucleus result from over-expression of this protein. These aggregates do not colocalize with IN-labeled VRCs. We prefer to show a representative track for a single IN spot, but below is the ensemble average plot requested by the reviewer obtained by averaging loss of PA-C6 from 4 nuclear IN spots.

10. Figure 6G. Please, explain why you observe dotted vDNA speckles in all samples treated with 25 μ M PF74. Are the cells viable upon 25 μ M PF74 treatment?

A: On a few occasions, we observed diffuse EdU/vDNA signal overlapping the nucleoli which was not colocalized with IN puncta. This background signal can be seen in all conditions

including DMSO (Fig. 7c) after 72 hpi or later. MDMs are viable even after 5 days of PF74 treatment, as shown in the new Suppl. Fig. 7.

11. Suppl. Fig. S3. Please, include representative images of the different treatment conditions of the Table. Especially for PF74 25 μ M.

A: Videos showing the effect of PF74 and Hexanediol treatment on the nuclear VRC clusters were included (Suppl. Video 3, Ex1 and Ex2). The images are not shown due to space limitation.

12. Please carefully review writing. According to Figure 6G, H, prolonged exposure to 25 μ M PF74 did NOT affect vDNA synthesis in the nucleus.

A: We thank the reviewer for catching this mistake, we corrected the sentence to say “did not affect vDNA synthesis in the nucleus”.

Reviewer #3:

1. A relatively high MOI is used, which seems consequent with respect to the clustering questions asked, however, my understanding from discussions with HIV experts is that a MOI of 1 is the most (if not only) relevant form of infection in human. Is there support for MOI's this high in macrophages? Could a bit more background be presented to position this choice? Or is the assumed scenario for this study such that the MOI is justified otherwise?

A: We would like to point out that we used higher MOIs (as determined in TZM-bl cells) merely to illustrate the tendency of VRCs to accumulate at distinct nuclear locations. In addition, it is difficult to determine true MOI in MDMs due to the extremely slow time course of infection. As shown in Suppl. Fig. 1, the nominal MOI of MDM infection after 5 days is much lower than MOI determined after 2 days in TZM-bl cells. We used an apparent MOI of 0.5 in live-cell imaging experiments and an apparent MOI of up to 5 (less than 10% MDMs infected, Suppl. Fig. 1d) for fixed MDM imaging. The MOI values used in the paper were determined in TZM-bl cells and the difference with MDMs is duly acknowledged in the main text and figure legends where appropriate.

2. The conclusion at the end of page 6 of having shown a “default pathway” seems a bit to big – it seems to me that more biochemical studies would be needed to establish such a fundamental statement across multiple cell types.

A: In this manuscript, we observed excellent localization of HIV-1 with nuclear speckles that was dependent on the CPSF6/CA interactions and a strong preference for integration into speckle associated genomic domains (SPADs). These effects were independent of cell type, leading to the conclusion that HIV-1 transport to NSs is a default pathway leading to HIV-1 integration site selection. Biochemical experiments are challenging and are unlikely to provide more reliable spatiotemporal information on compartmentalization compared to imaging.

3. For the PF74 experiment, what is brief? If I wanted to reproduce this experiment, I would have a hard time to guess what you did exactly and with what kind of variance.

A: We defined brief as a 30 minute treatment which was described in the text and figure legends. We now more clearly define brief incubation as a 30-minute treatment in the main text.

4. I do have some problems following the imaging protocols and analysis steps taken, and am missing significant information on image processing for the provided figures and supplemental material. I say so after spending quite some time trying to align the described z-stack time series with the presented time points in the movies.

A: We added pertinent information to Methods.

5. It would help if it could be indicated clearly which experiment was run on which microscope platform.

A: We started this project several years ago when we had a Zeiss LSM 780 microscope. The revised manuscript no longer contains images acquired using that microscope. All presented data sets were acquired using a Zeiss LSM 880 AiryScan microscope, as now stated in Methods.

6. From the provided information it is unclear to me what effective sampling size [nm per pixel] was used and if magnification was fixed to the optical mag of the objective or if additional settings were optimized.

A: Thank you for pointing this out. In the Methods section of revised manuscript, we provide additional information regarding the pixel size and optical zoom used for the distinct live-cell and fixed cell imaging modes. Briefly, all live-cell imaging was performed using 1x zoom and 240 nm/pixel with attenuated laser intensity to avoid photobleaching of single virus particles. In fixed cell mode, we have used a pixel sizes of 0.07-0.12 μm and a 2x optical zoom, 1024x1024 pixel images.

7. Laser power in % is not interpretable to me and is also only provided for the 405 nm line. What is the variance day to day? Over what time frame were experiments conducted, were detectors calibrated at all?

A: We have added additional information on the relative power for other 488, 561 and 633 laser lines. We used a standard Zeiss LSM 880 microscope that is regularly maintained and calibrated by service engineers. Day-to-day laser intensity variance is negligible, according to the manufacturer's specs. Zeiss does not provide us with the actual excitation light power at the specimen plane. The % laser intensities were adjusted empirically to achieve acceptable signal-background ratio while minimizing photobleaching.

8. I am still trying to make a best guess if a typical z-stack contained 45 planes or 45 z-stacks were taken,

A: A typical z-stack in live-cell imaging contained 11-15 z-planes spaced at 0.7-1 μm apart. For fixed cell imaging, 45 z-planes were acquired with 0.3 μm spacing. This information is now stated in Methods.

9. ...if 40-90 sec was the imaging duration of imaging frequency for stacks (respectively 5-7 minutes) why these speeds were sufficient to track particles (in multiple movies it is very hard to see any signal under the lines that were superimposed) and how 2.5 frames per second relate to 20s/ stack (different number

of planes?) and if the 20s are the imaging duration or time between imaging events.

A: Sorry for the confusion. We have now updated the Methods section to highlight different imaging modalities. We used 40-90s or 20s/frame when collecting 9-15 z-stack volumes. We used 2.5 s/frame when imaging a single z-plane to look at PA-C6 mobility after photoactivation. To track the PF74-induced increase in nuclear IN puncta mobility (shown in Fig. 5e, f and Suppl. Video 7) we used AiryScan-fast imaging mode (using 16-Airscan detectors) to image a single nucleus in a smaller frame size format. In this imaging mode, only the Hoechst-33342 (ex. 405nm) and INmNG (ex. 488nm) were imaged using the following parameters: 142 nm pixel size, Zoom 1.6x, pixel dwell time 0.73 μ s, Z-stacks spaced by 0.5 μ m and imaging frequency 2.5 s/volume for a period of 1 hour. Airyscan images were processed using a proprietary 3D-AiryScan processing module in Zen software using Auto-thresholding. The 3D images were later converted to 2D-maximum intensity projections for single particle tracking using ICY software. 3D-image series were analyzed off-line using ICY image analysis software (<http://icy.bioimageanalysis.org/>). This is now described in Methods.

10. Was the pixel dwell time generally 1.5 microseconds or just for one imaging condition?

A: Unless otherwise stated, the pixel dwell time was always 1.5 μ s, and bi-directional scanning was used. This information is provided in updated Methods.

11. Given the long time periods, if I understand correctly over 40 hours and more) it would be informative to know if lateral stage drift was assessed, or even corrected for and how.

A: An axial drift was corrected using the Z-piezo stage and DefiniteFocus2.0 module. This information is now included in Methods. We did not experience a significant lateral drift in our experiments.

12. Given the different N.A.s and refractive index mismatches, were R.I. of media controlled, what was the same resolution achieved on both systems? Was collection efficiency tested?

A: In this study, we have not pushed the limits of spatial resolution. PSF and chromatic aberrations for different objectives were tested using 150 nm Tetraspec beads on both 780 and 880 microscopes. Collection efficiency was not tested.

13. I am missing information on image registration. How were axial offsets measured and corrected, how large were they? Same question for lateral aberrations.

A: As stated above, the Z-Piezo stage and DefiniteFocus module (Carl Zeiss) were utilized to correct for axial drift. We have not measured axial offsets beyond ensuring acceptable voxel registration while imaging sub-resolution Tetrapeck beads in different colors.

14. What was the width chosen for the emission filter (if applicable) or the spectral detector settings?

A: Images were collected using GasP detectors with the following filter settings: Hoechst/AlexaFluor405 (415-470 nm), INmNG (490-558nm), CypA-DsRed/ AlexaFluor568 antibodies (572-625nm) and SNAP-SiR647/AlexaFluor647 antibodies/Cy5 conjugated antibodies (640-700nm). This information is now included in Methods.

15. What gain was used and how was the setting established? Was linearity checked?

A: The Gasp detector Gain setting was 750. This gain was determined empirically to achieve the best S/N ratio while reducing photobleaching.

16. In several movies the jump distance between presented time points was bigger than the diffraction limited, sometime substantially bigger, what is the proof (or rational) to assume the particle identity is known?

A: A sudden jump is caused by manual addition of drugs to the imaging chamber. Single particle motions were tracked pre- and post-drug addition. The relative positioning of single particles with respect to the lamin and between each other was used to manually confirm their identity before joining SPT tracks in the ICY track manager plugin.

17. My confusion for the image processing part is even bigger:

What was done during the 3D image processing? It is unclear if linear adjustments for presentation were made (okay to do if documented) and if the processing included any filtering (as the data presented might suggest, unless the code and compression caused a blur), was any registration between channels applied?

A: All images presented in the article were filtered using a smooth filter (strength 4) available on the Zen software (Zeiss). The same linear adjustments of grey values was applied for images presented side-by-side in different figures. Note, this routine was used only for image

presentation and not software assisted image analysis. Channel registration was not adjusted. This information is now added to Methods.

18. What are “2D single particle tracking parameters”? What distance and trace filling values were chosen, and why/based on which control?

A: Default SPT parameters in ICY were used to obtain particle trajectories. 2D objects were detected using a wavelet-based spot detector, and single particle tracking was performed using a respective ICY module. The spot tracking plugin uses default parameters estimated for diffusion or active transport. We used a mixed motion settings (diffusion/active transport) with no user-selectable tracking parameters. The quality of obtained tracks was verified by ensuring that these faithfully follow particle’s motion.

19. How is 5% “initial intensity” defined? Was background subtracted, was signal based on an area threshold or as integral from a multiparameter fit? If so, was the fit width consistent with the imaging parameters? What fit model was used? Was bleach correction applied, if so, based on which model, if not how was the signal interpretation adjusted to distinguish bleaching from particle decay?

A: The initial fluorescence signal at the time of imaging acquisition was set at 100% and used as “initial intensity”. The local background was subtracted in the ICY track manager using a radius of 3 pixels around an object. No models or fitting parameters or bleach correction was used. We do regularly control for photobleaching in our experiments to ensure <10% loss of signal by the end of image acquisition.

20. What was the criteria for a signal to be interpreted as single core? For CA/p24 some imaging parameters are given in the analysis section, but the time information is missing.

A: Isolated single viral particles adhered to coverslips were used as reference for cytoplasmic and nuclear HIV-1 complexes. Except for the nuclear complexes and result from merger of multiple VRCs, single punctate trafficking in curvilinear motion pattern in the cytoplasm and early after nuclear import most likely represent single virus complex. Viral clusters observed in the nuclei under relatively high MOI are identified based on their comparatively greater intensity relative to single intact virions. Time information is given in figure legends as hours post-infection (hpi).

21. Lamin signals are frequently not continuous and somewhat blurred. How was that handled in the analysis? Was dilation used generally and always with the same parameters? How many rounds were applied?

A: The lamin signal is often discontinuous due to insufficient axial sampling in live cell imaging settings. Single particles entering through poorly defined lamin boundaries were not analyzed. When imaging fixed cells, the lamin signal was robust due to higher laser power and better spatial sampling, allowing for reliable identification of the lamin signal by the HK-means plugin in ICY software. There is very little room for manipulation of HK-means detection and therefore we used same parameters across the board. Dilation of lamin signal in fixed cell by 1 pixel was used across the board (1 iteration) to define the nuclear borders.

22. Were SC35 compartments really varying in size between 10 and 2000 pixel? And were pixels of the same size?

A: The size/shape of SC35+ compartments varied broadly across cell types. The thresholds were manually adjusted and cross-verified for reliable detection of the SC35-immunostained compartments. The pixel size (0.07 or 0.14 μm) and image acquisition parameters (fixed cell imaging settings) were always the same across multiple cell types.

23. How were thresholds established? Did the data show plateaus (settings were changing the thresholding parameter slightly had no or little impact on the results)?

A: Our imaging parameters (live cell and fixed cell settings) were set to acquire only 1/4th of the grey scale values and never saturated. Thresholds were adjusted for a given cell type based on visual inspection of 3D rendered images and used consistently for all conditions. Varying a threshold within a reasonable range had no impact on the results, as far as the quantification of IN-associated spots is concerned. The caveats of this thresholding in estimating the size and number of individual speckles is acknowledged in Methods:

“Note: This type of speckle analysis occasionally picked multiple closely located speckles as one in all cell types except MDMs where NSs were well-separated. Because of this effect, the number of NSs in non-MDM cells is underestimated (Fig. 3c), while the individual speckle volumes are overestimated (Fig. 3d)”.

24. As a summary, I am not able to reproduce any of your analysis, you are not giving enough information.

A: We hope that additional details related to methods and image analyses provided in the revised manuscript will allow to reproduce the presented data.

25. The supplemental movies are very hard to interpret, OME-Tiff (or even Tiff) images would help as it seems you combined still images and used the copy number of the images to impact the play time of the movie. Extracting the stills from the movies I have trouble seeing the signal in some cases, but it is possible that a faint signal would be lost in the diverse format conversions. In general a collage of images that document a trace (with the z-value being indicated if 3D tracking is done) would be very helpful.

A: We would like to point out to the reviewer that collages of images for the movies are shown in the corresponding figures. As mentioned in Methods, only 2D tracking was performed.

26. On the statistical analysis, was the Mann-Whitney test the only applicable? Why would a two-sided ANOVA not fit your data?

A: We have used pairwise analysis of data and hence selected Mann-Whitney non-parametric test as an acceptable test.

References

1. Mamede, J.I., Cianci, G.C., Anderson, M.R. & Hope, T.J. Early cytoplasmic uncoating is associated with infectivity of HIV-1. *Proceedings of the National Academy of Sciences of the United States of America* **114**, E7169-E7178 (2017).

2. Francis, A.C. & Melikyan, G.B. Single HIV-1 Imaging Reveals Progression of Infection through CA-Dependent Steps of Docking at the Nuclear Pore, Uncoating, and Nuclear Transport. *Cell host & microbe* **23**, 536-548 e536 (2018).
3. Chin, C.R. *et al.* Direct Visualization of HIV-1 Replication Intermediates Shows that Capsid and CPSF6 Modulate HIV-1 Intra-nuclear Invasion and Integration. *Cell reports* **13**, 1717-1731 (2015).
4. Hulme, A.E., Kelley, Z., Foley, D. & Hope, T.J. Complementary assays reveal a low level of CA associated with nuclear HIV-1 viral complexes. *J Virol* (2015).
5. Peng, K. *et al.* Quantitative microscopy of functional HIV post-entry complexes reveals association of replication with the viral capsid. *eLife* **3**, e04114 (2014).
6. Bejarano, D.A. *et al.* HIV-1 nuclear import in macrophages is regulated by CPSF6-capsid interactions at the Nuclear Pore Complex. *eLife* **8** (2019).
7. Stultz, R.D., Cenker, J.J. & McDonald, D. Imaging HIV-1 Genomic DNA from Entry through Productive Infection. *J Virol* **91** (2017).
8. Marquez, C.L. *et al.* Kinetics of HIV-1 capsid uncoating revealed by single-molecule analysis. *eLife* **7** (2018).
9. Francis, A.C., Marin, M., Shi, J., Aiken, C. & Melikyan, G.B. Time-Resolved Imaging of Single HIV-1 Uncoating In Vitro and in Living Cells. *PLoS Pathog* **12**, e1005709 (2016).
10. Xu, J.P. *et al.* Exploring Modifications of an HIV-1 Capsid Inhibitor: Design, Synthesis, and Mechanism of Action. *J Drug Des Res* **5** (2018).
11. Burdick, R.C. *et al.* HIV-1 uncoats in the nucleus near sites of integration. *Proceedings of the National Academy of Sciences of the United States of America* **117**, 5486-5493 (2020).
12. Hulme, A.E., Kelley, Z., Foley, D. & Hope, T.J. Complementary Assays Reveal a Low Level of CA Associated with Viral Complexes in the Nuclei of HIV-1-Infected Cells. *J Virol* **89**, 5350-5361 (2015).
13. Burdick, R.C. *et al.* Dynamics and regulation of nuclear import and nuclear movements of HIV-1 complexes. *PLoS Pathog* **13**, e1006570 (2017).
14. Burdick, R.C., Hu, W.S. & Pathak, V.K. Nuclear import of APOBEC3F-labeled HIV-1 preintegration complexes. *Proceedings of the National Academy of Sciences of the United States of America* **110**, E4780-4789 (2013).
15. Zila, V., Muller, T.G., Laketa, V., Muller, B. & Krausslich, H.G. Analysis of CA Content and CPSF6 Dependence of Early HIV-1 Replication Complexes in SupT1-R5 Cells. *mBio* **10** (2019).
16. Pezeshkian, N., Groves, N.S. & van Engelenburg, S.B. Single-molecule imaging of HIV-1 envelope glycoprotein dynamics and Gag lattice association exposes determinants responsible for virus incorporation. *Proceedings of the National Academy of Sciences of the United States of America* **116**, 25269-25277 (2019).
17. Achuthan, V. *et al.* Capsid-CPSF6 Interaction Licenses Nuclear HIV-1 Trafficking to Sites of Viral DNA Integration. *Cell host & microbe* **24**, 392-404 e398 (2018).
18. Sowd, G.A. *et al.* A critical role for alternative polyadenylation factor CPSF6 in targeting HIV-1 integration to transcriptionally active chromatin. *Proceedings of the National Academy of Sciences of the United States of America* **113**, E1054-1063 (2016).
19. Saito, A. *et al.* Roles of capsid-interacting host factors in multimodal inhibition of HIV-1 by PF74. *J Virol* (2016).
20. Chen, Y. *et al.* Mapping 3D genome organization relative to nuclear compartments using TSA-Seq as a cytological ruler. *The Journal of cell biology* **217**, 4025-4048 (2018).
21. Langmead, B. & Salzberg, S.L. Fast gapped-read alignment with Bowtie 2. *Nat Methods* **9**, 357-359 (2012).
22. Khan, A. & Zhang, X. dbSUPER: a database of super-enhancers in mouse and human genome. *Nucleic Acids Res* **44**, D164-171 (2016).

23. Saito, A. *et al.* Capsid-CPSF6 Interaction Is Dispensable for HIV-1 Replication in Primary Cells but Is Selected during Virus Passage In Vivo. *J Virol* **90**, 6918-6935 (2016).
24. Matreyek, K.A. *et al.* Host and viral determinants for MxB restriction of HIV-1 infection. *Retrovirology* **11**, 90 (2014).
25. Serrao, E., Cherepanov, P. & Engelman, A.N. Amplification, Next-generation Sequencing, and Genomic DNA Mapping of Retroviral Integration Sites. *Journal of visualized experiments : JoVE* (2016).
26. Kent, W.J. BLAT--the BLAST-like alignment tool. *Genome Res* **12**, 656-664 (2002).
27. Kim, D., Langmead, B. & Salzberg, S.L. HISAT: a fast spliced aligner with low memory requirements. *Nat Methods* **12**, 357-360 (2015).
28. Quinlan, A.R. & Hall, I.M. BEDTools: a flexible suite of utilities for comparing genomic features. *Bioinformatics* **26**, 841-842 (2010).
29. Fei, J. *et al.* Quantitative analysis of multilayer organization of proteins and RNA in nuclear speckles at super resolution. *J Cell Sci* **130**, 4180-4192 (2017).
30. Kim, B., Nguyen, L.A., Daddacha, W. & Hollenbaugh, J.A. Tight interplay among SAMHD1 protein level, cellular dNTP levels, and HIV-1 proviral DNA synthesis kinetics in human primary monocyte-derived macrophages. *The Journal of biological chemistry* **287**, 21570-21574 (2012).

Reviewers' Comments:

Reviewer #1:

Remarks to the Author:

This manuscript by Answath and coworkers is improved from the prior submission. This revision focuses more on the primary points of interest noted by myself and other reviewers, namely 1) The accumulation of HIV VRCs in the nucleus of macrophages, which the authors call HIV-1 clusters 2) These cluster accumulate in nuclear speckles in a CA dependent fashion. By getting right to these points in MDMs and then using other cell lines to demonstrate this is not unique to MDMs, the flow of the manuscript is now logical and straightforward. The authors have de-emphasized the areas of disagreement with other studies in favor of the main story, and approach this discord in the discussion rather than letting it spill into, and dominate, the results section. There remains some wording issues that I noted that I would encourage but not insist the authors to change, as noted below, but generally speaking I think this is a strong study that warrants publication in Nature Communications.

Minor editorial details:

Remove strikingly from the abstract.

Fig 2 D is cropped in a way that removes text from the figure in the version I reviewed.

The legend of figure 2 seems to swap panels C and D.

The data in figures 2A-D could be clarified, as I think it is important. Specifically, I would recommend clarifying the colocalization analysis provided in the graph. It is awkwardly worded in the legend/text. This is an interesting experiment and outcome, explaining it more methodically seems worth the effort. I am concerned that I understand it because I have seen the data presented and I regularly perform imaging experiments such as these but the average reader may not appreciate the interesting nature of this result.

Drastic in line 338 of discussion could be replaced with a more dispassionate adjective.

Best,
Ed Campbell

Reviewer #2:

Remarks to the Author:

This is a minimally responsive review in which the authors argue a lot but do not provide substantial new data. As such, the manuscript remains largely descriptive and lacks functional justification for the conclusions drawn. The A77V mutant which provided some functional validation is now removed from the current manuscript and also does not affect viral replication in macrophages, CD4 T cells and humanized mice. It is unclear why it was included, only to provide evidence that the CPSF6-CA interaction is not relevant for viral replication? This and the fact that all 3 reviewers were confused with the description of the methods and results throughout the manuscript, the paper appears more suitable for a specialized journal.

We thank the reviewers for going over the revised manuscript and their comments. Below are point-by-point responses to their concerns.

REVIEWERS' COMMENTS:

Reviewer #1 (Remarks to the Author):

This manuscript by Answath and coworkers is improved from the prior submission. This revision focuses more on the primary points of interest noted by myself and other reviewers, namely 1) The accumulation of HIV VRCs in the nucleus of macrophages, which the authors call HIV-1 clusters 2) These cluster accumulate in nuclear speckles in a CA dependent fashion. By getting right to these points in MDMs and then using other cell lines to demonstrate this is not unique to MDMs, the flow of the manuscript is now logical and straightforward. The authors have de-emphasized the areas of disagreement with other studies in favor of the main story, and approach this discord in the discussion rather than letting it spill into, and dominate, the results section. There remains some wording issues that I noted that I would encourage but not insist the authors to change, as noted below, but generally speaking I think this is a strong study that warrants publication in Nature Communications.

Minor editorial details:

- Remove strikingly from the abstract.
A: Removed.
- Fig 2 D is cropped in a way that removes text from the figure in the version I reviewed.
A: Fig. 2D is now intact
- The legend of figure 2 seems to swap panels C and D.
A: Corrected, thank you.
- The data in figures 2A-D could be clarified, as I think it is important. Specifically, I would recommend clarifying the colocalization analysis provided in the graph. It is awkwardly worded in the legend/text. This is an interesting experiment and outcome, explaining it more methodically seems worth the effort. I am concerned that I understand it because I have seen the data presented and I regularly perform imaging experiments such as these but the average reader may not appreciate the interesting nature of this result.

A: We thank the reviewer for this suggestion. The legend is modified as follows:

Fig. 2. Multiple HIV-1 VRCs merge in nuclear speckles of MDMs. (a, b) MDMs were co-infected with two VSV-G/HIV-eGFP pseudoviruses labeled with INmNG or INmCherry (MOI 2), fixed at 6 hpi and immunostained for NSs (SC35). (a) A central section of MDM nucleus shows that merger of INmNG and INmCherry VRCs occur in NSs, as evidenced by colocalization of double positive IN clusters (yellow arrows) with SC35 staining. Single-labeled INmNG and INmCherry VRCs *en route* to forming clusters are marked with dashed green and red circles, respectively. The contours of NSs are marked with semi-transparent grey dashed lines. (b) The fluorescence intensities associated with INmNG VRC puncta inside (SC35(+)) or outside of NSs (SC35(-)) are plotted. A proportional increase of INmCherry signals with INmNG signals detected in

SC35(+) compartments is in contrast to VRCs in SC35(-) nucleoplasm, revealing a clear tendency for VRCs to cluster in NSs.

- Drastic in line 338 of discussion could be replaced with a more dispassionate adjective.

A: Replaced with “marked”

Reviewer #2 (Remarks to the Author):

This is a minimally responsive review in which the authors argue a lot but do not provide substantial new data. As such, the manuscript remains largely descriptive and lacks functional justification for the conclusions drawn.

The A77V mutant which provided some functional validation is now removed from the current manuscript and also does not affect viral replication in macrophages, CD4 T cells and humanized mice. It is unclear why it was included, only to provide evidence that the CPSF6-CA interaction is not relevant for viral replication?

A: The A77V data are presented in Fig. 2 and have not been removed from the paper. The original manuscript analyzed A77V EdU/vDNA signals and its colocalization with nuclear speckles, whereas the current version shows analysis of A77V VRCs (IN labeled), as per the Reviewer 1's request to stick to the analysis of VRCs, instead of EdU spots. The outcome of both analyses is the same. We would like to point out that the A77V control was used only to demonstrate the CPSF6 dependence of nuclear transport to speckles. We do not make claims about the requirement of this interaction for HIV-1 replication, as HIV-1 can integrate into lamin-associated genes and replicate in the absence of CPSF6.

Please see Discussion: “The mechanism of stable HIV-1 cluster formation in MDM nuclei and its role in infection are unknown. Given that the A77V/CA mutant virus, which is able to replicate in MDMs, does not reach NSs or form clusters (Fig. 2c-f and Supplementary Fig. 4c, e), VRC transport to and accumulation in NSs appear largely dispensable for HIV-1 infection. However, given the strong propensity for HIV-1 to integrate into SPADs (Fig. 8), the localization of VRCs in NSs is functionally conserved. Accordingly, in growth competition experiments, WT HIV-1 that can bind CPSF6 consistently outpaces the A77V mutant virus that is defective for CPSF6 binding³⁵.”